# 3D-RAD: A Comprehensive 3D Radiology Med-VQA Dataset with Multi-Temporal Analysis and Diverse Diagnostic Tasks

**Xiaotang Gai[1,2],[\*], Jiaxiang Liu[1,3],[\*], Yichen Li[1,2],[\*], Zijie Meng[1,2], Jian Wu[2], Zuozhu Liu[1,2],[†]**

[1]ZJU-Angelalign R&D Center for Intelligence Healthcare, Zhejiang University, China
[2]Zhejiang Key Laboratory of Medical Imaging Artificial Intelligence, Zhejiang University, China
[3] Guangdong Institute of Intelligence Science and Technology, Hengqin, Zhuhai, China
{xiaotang.23, zuozhuliu}@intl.zju.edu.cn

 **Code:** https://github.com/Tang-xiaoxiao/3D-RAD
 **Dataset:** https://huggingface.co/datasets/Tang-xiaoxiao/3D-RAD

## Abstract

Medical Visual Question Answering (Med-VQA) holds significant potential for clinical decision support, yet existing efforts primarily focus on 2D imaging with limited task diversity. This paper presents 3D-RAD, a large-scale dataset designed to advance 3D Med-VQA using radiology CT scans. The 3D-RAD dataset encompasses six diverse VQA tasks: anomaly detection, image observation, medical computation, existence detection, static temporal diagnosis, and longitudinal temporal diagnosis. It supports both open- and closed-ended questions while introducing complex reasoning challenges, including computational tasks and multi-stage temporal analysis, to enable comprehensive benchmarking. Extensive evaluations demonstrate that existing vision-language models (VLMs), especially medical VLMs exhibit limited generalization, particularly in multi-temporal tasks, underscoring the challenges of real-world 3D diagnostic reasoning. To drive future advancements, we release a high-quality training set 3D-RAD-T of 136,195 expert-aligned samples, showing that fine-tuning on this dataset could significantly enhance model performance. Our dataset and code, aiming to catalyze multimodal medical AI research and establish a robust foundation for 3D medical visual understanding, are publicly available.

## 1 Introduction

Recent advances in Medical Visual Question Answering (Med-VQA) show strong potential for clinical diagnosis support [1–6]. By integrating medical imaging and natural language processing, Med-VQA bridges multimodal analysis and clinical reasoning, enabling scalable solutions for automated interpretation, physician assistance, and patient education. Moreover, the evolution of these datasets highlights both progress and ongoing challenges in the medical domain[7–10].

The field was initially constrained by small-scale datasets with limited diversity. The VQA-RAD dataset[11], introduced through the ImageCLEF Med-VQA challenges, pioneered structured clinical questions across four categories: imaging modality, anatomical plane, organ system, and abnormalities. Subsequent iterations (e.g., VQA-med-2019 to VQA-med-2021) [12] expanded the data volume and refined medical taxonomies, but remained restricted to narrow clinical scopes. Concurrently, SLAKE (2021) [13] emerged as the first bilingual (English-Chinese) dataset, incorporating 642

---

[\*]Equal contribution.
[†]Corresponding author.

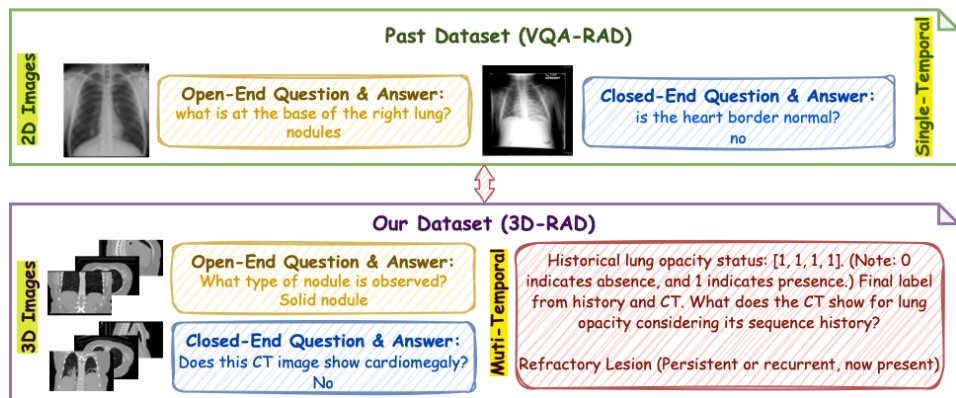

Figure 1: Qualitative Comparison Across Dataset Types. Prior work (e.g., VQA-RAD; top) focuses on 2D open- and closed-ended VQA, whereas our 3D-RAD (bottom) additionally includes 3D imaging and multi-temporal tasks.

images and 14,028 QA pairs annotated with semantic labels to enhance reasoning. Meanwhile, PathVQA (2022) [14] addressed a critical gap in histopathology by curating 32,000 whole-slide images with 1.2 million QA pairs, enabling models to interpret fine-grained cellular patterns and tumor grading criteria. Despite these advances, PMC-VQA (2023) [15] further demonstrated the challenges of scaling specialized domains, integrating over 200,000 pathology images with hierarchical question frameworks spanning cancer subtyping and prognosis prediction. While these datasets established strong baseline benchmarks for radiology, pathology, or multilingual scenarios, their limited scale (e.g., SLAKE's 642 images) and task specificity (e.g., PathVQA's focus on single-modality histopathology) hindered generalization to complex clinical workflows requiring cross-modal reasoning and multi-disciplinary knowledge integration [16–18].

Advancing existing Med-VQA datasets for real-world clinical applications faces three key challenges. First, most benchmarks are curated with 2D medical images or 2D slices derived from 3D scans, lacking critical volumetric details inherent in 3D imaging [1, 19–23]. In clinical practice, accurate diagnosis and treatment planning for many conditions depend on a precise understanding of 3D spatial relationships and correlations, as seen in CT or MRI scans [16]. Second, current VQA tasks, such as multiple-choice questions or brief open-ended responses with 3-5 words, are often simplified to facilitate rule-based evaluation. These tasks rarely address complex, real-world scenarios like medical computations, quantitative measurements, or temporal analysis involving historical records [24, 25, 13]. Finally, the scale and granularity of existing datasets limit comprehensive disease analysis. Moreover, apart from robust evaluation of 3D medical image understanding, a large-scale, high-quality training set is also essential to significantly enhance the performance of current medical vision-language models (Med-VLMs) for 3D diagnostic tasks [16, 15, 26–28].

In this paper, we introduce 3D-RAD, a large-scale dataset designed for effective training and comprehensive benchmarking of 3D Med-VQA tasks using radiology CT and text. The 3D-RAD consists of 170K data entries across six challenging VQA task types—spanning open-ended (e.g., anomaly detection, image observation, medical computation) and closed-ended (e.g., existence detection, static temporal diagnosis, longitudinal temporal diagnosis). It covers over 18 diseases and supports advanced reasoning tasks, such as medical computation and temporal analysis, making it highly suitable for practical evaluation (see Figure 2). The dataset adheres to data usage regulations and ethical considerations. We leverage strong large language models (LLMs) for annotation, with rigorous consistency checks across LLMs to ensure high-quality annotations, supplemented by extensive human validation for quality control. Additionally, we provide a training subset, 3D-RAD-T, with fine-tuned models demonstrating substantial performance gains. By releasing 3D-RAD, we aim to provide a reliable resource to advance 3D Med-VQA, facilitating research into network architectures and pretraining strategies optimized for 3D multimodal clinical reasoning. Our main contributions are summarized as follows:

- We propose the first large-scale benchmark explicitly designed for 3D Med-VQA, featuring multi-temporal and multi-task settings on volumetric CT data. Unlike prior benchmarks focused on 2D or limited tasks, our benchmark fully leverages the 3D nature of radiology data to evaluate the comprehensive capabilities of vision-language models.

Table 1: Comparison of 3D-RAD with Existing Med-VQA Datasets across Modality, Scale, Tasks Covered, Temporal Reasoning, 3D Support, and Quality Check.

| Dataset | Modality | Dataset Scale | Tasks Covered | Temporal Reasoning | 3D Support | Quality Check |
|---|---|---|---|---|---|---|
| **VQA-RAD** [11] | 2D (X-ray, CT) | 315 images 3,515 QA | Modality, anatomy, abnormalities | ✗ | ✗ | Dual Annotation |
| **SLAKE** [13] | 2D | 642 images 14,028 QA | Function, anatomy | ✗ | ✗ | - |
| **PathVQA** [14] | 2D (Pathology) | 4,998 images 32,799 QA | Histopathology | ✗ | ✗ | Manual Validation |
| **VQA-Med** [12] | 2D | ∼5K images | Classification, description | ✗ | ✗ | Manual Validation |
| **M3D-VQA** [16] | 3D (CT) | 120K QA | Slice finding, spatial reasoning, diagnosis | ✗ | ✓ | - |
| **3D-RAD (Ours)** | 3D (CT) | **16,188** images, **170K** QA across 6 tasks | Detection, quantification, diagnosis, **progression** | ✓ | ✓ | LLM Score + **Human** Validation + **Consistency Check** |

- Through extensive experiments and analyses across carefully designed evaluation dimensions—anomaly detection, image observation, medical computation, existence detection, static temporal diagnosis, and longitudinal temporal diagnosis—we reveal critical limitations in current 3D vision-language models. In particular, our results expose significant performance gaps in complex multi-temporal reasoning tasks, highlighting the huge space for improvement.

- Based on these findings, we release a high-quality training dataset of **136K** samples and fine-tune state-of-the-art 3D vision-language models. We also demonstrate that domain-specific 3D training yields substantial performance improvements, providing a valuable resource to advance research in 3D medical VQA.

## 2 Related Work

### 2.1 2D Med-VQA Datasets

Med-VQA, positioned at the intersection of medical imaging and natural language processing, has historically focused on 2D static images. Early datasets like VQA-RAD (2018) [13] introduced structured clinical QA tasks but were limited in image scale and disease coverage. PathVQA (2020) expanded into pathology with fine-grained tasks, yet relied heavily on binary questions. SLAKE (2021) [13] added bilingual annotations and knowledge graph support but was constrained by a small dataset size. The VQA-Med series (2018–2021) broadened task types yet remained centered on basic radiology, lacking support for high-level decision-making tasks [2, 29, 3].

Overall, current 2D Med-VQA datasets face three core challenges: (1) limited spatial modeling, hindering representation of lesion progression or dynamic anatomy; [11] (2) insufficient support for clinical decision-making, with an emphasis on basic recognition tasks [13]; and (3) poor generalizability to real-world, multi-modal clinical workflows. These gaps restrict model deployment in complex clinical contexts [16]. Therefore, there is a pressing need for a large-scale 3D Med-VQA benchmark that supports multi-temporal reasoning, spatial understanding, and clinical interpretability, advancing the field from perception to real-world diagnostic reasoning [30].

### 2.2 3D Med-VQA Datasets

With the growing adoption of volumetric imaging and intraoperative navigation [8, 6], 3D Med-VQA is becoming essential for advancing intelligent clinical decision support [16]. Unlike traditional 2D datasets, 3D VQA enables rich spatial and temporal reasoning aligned with real-world diagnostic workflows. Recent efforts such as M3D-VQA (2023) [16] and RadFM [31] have laid foundational work—M3D-VQA with its 120K CT-based questions across five clinical tasks, and RadFM as a unified 2D/3D medical foundation model. However, existing datasets remain limited in scale, task diversity, and temporal modeling, hindering comprehensive evaluation and progress [16].

To bridge this gap, we introduce 3D-RAD, a large-scale and clinically grounded 3D Med-VQA benchmark. It defines six task types spanning recognition, detection, diagnosis, anatomical reasoning, and multi-temporal progression. Unlike prior datasets focused on static or single-task settings, 3D-

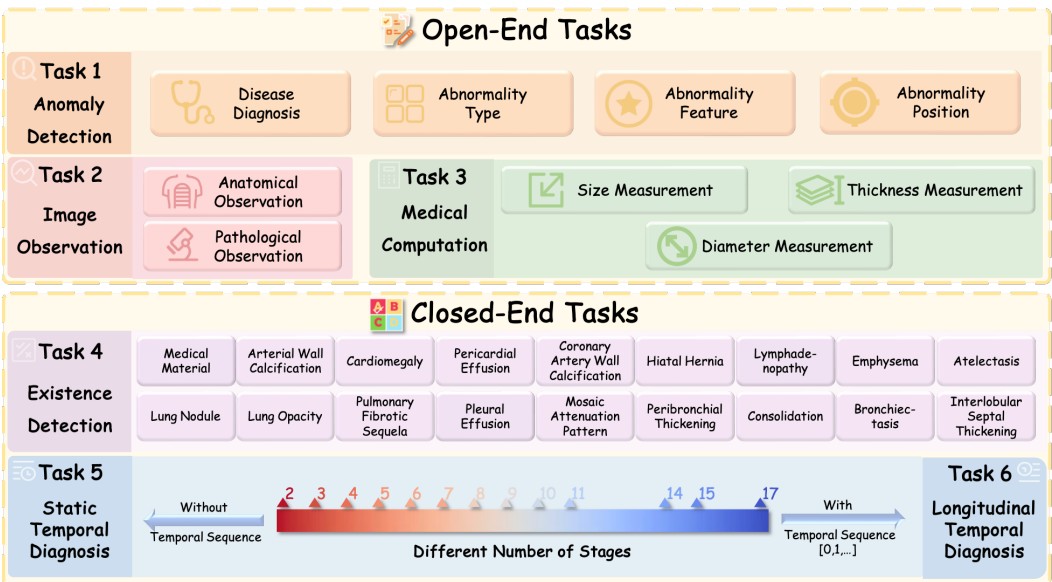

Figure 2: Definitions of Open-Ended and Closed-Ended Tasks in 3D-RAD. Different colors indicate distinct tasks; items sharing the same color represent different subtasks within that task.

RAD supports dynamic, longitudinal reasoning by incorporating follow-up imaging and historical reports. It provides high-quality annotations, expert-validated QA, and systematic evaluation across multiple VLMs. Extensive experiments demonstrate current models struggle with complex 3D and temporal tasks, especially multi-temporal reasoning. 3D-RAD thus offers a scalable, extensible platform to advance multimodal reasoning, fairness, and clinical applicability in 3D Med-VQA.

## 3    3D-RAD DATASET

In this section, we introduce the task definitions and dataset construction process of the 3D-RAD dataset, as illustrated in Figure 2 and Figure 3. Our goal is to build a comprehensive 3D medical visual question answering benchmark that reflects real-world clinical reasoning demands across multiple diagnostic scenarios[32]. To this end, we first analyze large-scale radiology CT scans and their associated reports to identify common disease categories, anatomical structures, and clinical question types. Based on these insights, we define 6 representative VQA tasks covering both static and dynamic reasoning needs: *1). Anomaly detection, 2). Image Observation, 3).Medical Computation, 4). Existence Detection, 5). Static Temporal Diagnosis,* and *6). Longitudinal Temporal Diagnosis.*

The dataset construction process involves a semi-automated pipeline combining radiology report parsing, expert annotation, and temporal case matching. We further incorporate both closed-ended and open-ended questions to capture different forms of clinical reasoning, and ensure label quality through multi-temporal validation. The final dataset comprises samples, each paired with 3D CT scans, providing a diverse and challenging benchmark for advancing 3D Med-VQA research. We partition the dataset into two distinct subsets to ensure strict separation between training and evaluation, preventing data leakage and enabling fair, reproducible comparisons.

- **3D-RAD-Bench**: approximately 34K QA pairs with 2,662 images across six representative tasks, designed for standardized evaluation)
- **3D-RAD-T (Train)**: approximately 136K QA pairs with 13,526 images constructed independently from the benchmark using the same procedure, with no overlap, expansion, or resampling.

### 3.1    Tasks Definitions

**Task 1: Anomaly Detection** is essential in medical imaging for uncovering abnormal patterns that may signify rare or critical conditions[33]. The model must detect deviations from normal anatomical structures, specifying the type, characteristics, and location of anomalies. This task is closely related to disease diagnosis, particularly in recognizing hallmark pathological findings. For example, in

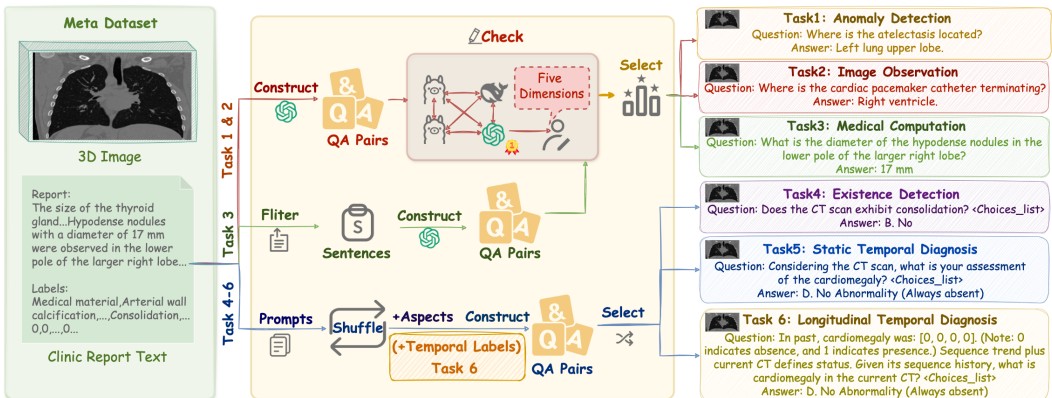

Figure 3: 3D-RAD Dataset Construction Pipeline. Left: meta dataset with a 3D scan, clinical report, and structured labels. Middle: QA construction—open-ended (Tasks 1–2) from selected report sentences with a five-dimension quality check; numeric (Task 3) from measurements; closed-ended (Tasks 4–6) via prompt templates and choice lists, with temporal indices for Task 6. Right: representative QA examples for Tasks 1–6.

response to the question "Where is the consolidative parenchyma area located?", the answer could be "left lower lobe of the left lung". It includes four subtasks: Disease Diagnosis: Identifies the main clinical abnormality; Abnormality Type: Categorizes the nature of the abnormal finding (e.g., mass, effusion); Abnormality Feature: Describes visual or structural characteristics of the abnormality (e.g., spiculated, cavitated); Abnormality Position: Specifies the anatomical location of the detected abnormal region.

**Task 2: Image Observation** in the medical domain remains highly challenging due to the heterogeneity of imaging modalities and their fundamental divergence from natural images[34]. This task focuses on analyzing and extracting descriptive information from medical images. Specifically, it aims to identify both anatomical and pathological observations. This task evaluates a model's basic perceptual capabilities in understanding medical images. For example, in response to "What type of opacities are observed?", a answer would be "ground-glass opacities". Unlike Task 1, which focuses on abnormal findings, Task 2 also includes observations of normal structures (e.g., heart stents).

**Task 3: Medical Computation** addresses numerical reasoning under label-scarce settings, enabling broader generalization across diseases and patient populations[35]. This task evaluates the model's ability to perform quantitative reasoning and measurements based on 3D medical image data. The task involves extracting and computing values such as Size, Diameter and Thickness. While some large models (e.g., ChatGPT) demonstrate preliminary abilities in medical computation. For example, in response to "What is the diameter of the largest nodule in the left lower lobe superior?", the correct answer may be "5 mm", yet the model might overestimate it as "Greater than 10 mm".

**Task 4: Existence Detection** aims to assess the presence or absence of clinical findings in 3D scans, addressing the domain gap that limits the effectiveness of vision-language models trained on natural images[36]. This is a binary classification task designed to assess whether specific abnormalities or pathologies are present in the image. For each input image, the model must predict yes or no for a set of 18 predefined aspects. These aspects cover a broad range of abnormalities, allowing us to evaluate model performance across diverse pathological categories. Some models perform well on specific aspects but show marked performance drops on others, revealing limitations in generalization.

**Task 5: Static Temporal Diagnosis** is a novel task introduced in 3D-RAD to evaluate a model's ability to infer temporal lesion status based solely on a single current 3D scan. Although diseases often evolve over time, this task does not provide prior image or label history. Instead, models must classify each of the 18 diagnostic labels into one of four temporal categories by recognizing spatial and morphological cues indicative of lesion progression: 1. *Refractory Lesion* (persistent or recurrent, now present), 2. *Resolved Lesion* (previously present or recurrent, now absent), 3. *New Lesion* (absent previously, now present), 4. *No Abnormality* (always absent). This setting reflects a realistic diagnostic challenge where temporal reasoning must be inferred implicitly from static image features. Note that in both Task 5 and Task 6, the subtask indices correspond to different scan counts available for each case.

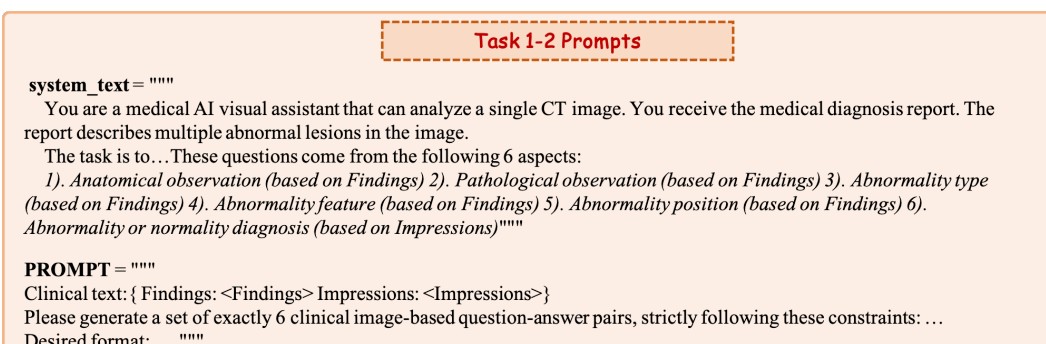

Figure 4: A Concise Prompt Example for Tasks 1–2. See Figure 11 for the full prompt.

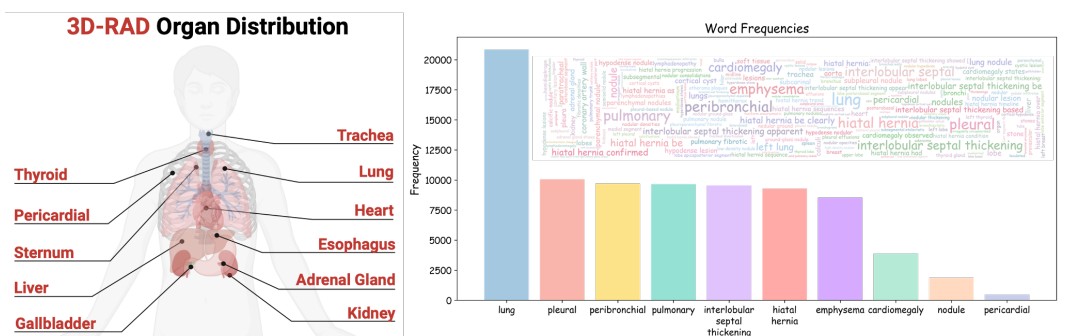

Figure 5: Data Distribution of 3D-RAD Dataset. Left: Organ distribution derived from named-entity recognition (NER) over all 3D-RAD QA pairs, showing coverage of major thoracic and upper-abdominal structures. Right: QA-based NER term statistics—bar chart of the most frequent entities/concepts alongside a word cloud that highlights the broader long-tail vocabulary of anatomical and pathological descriptors.

**Task 6: Longitudinal Temporal Diagnosis** extends Task 5 by introducing explicit historical diagnostic context to support more accurate temporal reasoning. As current VLMs cannot directly process multiple 3D scans, prior information is provided as a sequence of binary labels indicating lesion presence (1) or absence (0) over time. Given this history and the current scan, models are required to classify the lesion status into the same four temporal categories as in Task 5. For example: *Arterial wall calcification progression: [1, 0]. Q: Based on the sequence and current scan, what best describes arterial wall calcification?* By incorporating longitudinal cues, this task enables more informed decisions in challenging cases such as recurrent or resolving lesions. It thus evaluates the model's ability to integrate temporal context into diagnostic inference.

Tasks 5 and 6 are designed to address a core clinical challenge: monitoring lesion evolution over time through follow-up imaging. In practice, assessing treatment response relies on identifying temporal lesion status—whether a lesion is new, resolved, persistent, or absent—often requiring detailed anatomical alignment across scans. However, this manual process is time-consuming and error-prone. Despite the temporal nature of follow-up diagnosis, most existing deep learning models process each scan independently, without modeling progression[37, 38]. By formulating temporal lesion classification under both static (Task 5) and longitudinal (Task 6) conditions, our dataset provides a systematic benchmark for evaluating a model's capacity for implicit and explicit temporal reasoning in 3D medical imaging.

## 3.2 Dataset Construction Pipeline

### 3.2.1 Dataset Construction

*Data Source:* 3D-RAD is built upon CT-RATE [30], a large-scale 3D chest CT dataset paired with radiology reports, licensed under CC BY-NC-SA. We collect 16,188 CT scans from 11,255 patients, strictly separating training and benchmark sets by following the original training/validation split. For

Table 2: Distributions of Images, Patients, and QA pairs across tasks in the 3D-RAD training and benchmark sets.

| Source | Category | Task 1 | Task 2 | Task 3 | Task 4 | Task 5 | Task 6 | Overall |
|---|---|---|---|---|---|---|---|---|
| 3D-RAD-Bench | Images | 1,858 | 980 | 656 | 1,304 | 169 | 169 | 2,662 |
| | Patients | 1,008 | 651 | 331 | 1,304 | 169 | 169 | 1,304 |
| | Q-A Pairs | 2,666 | 1,024 | 1,002 | 23,472 | 2,873 | 2,873 | 33,910 |
| 3D-RAD-T (Train) | Images | 5,670 | 2,045 | 1,148 | 5,565 | 774 | 774 | 13,526 |
| | Patients | 4,443 | 1,697 | 587 | 5,565 | 774 | 774 | 9,951 |
| | Q-A Pairs | 6,055 | 2,081 | 1,573 | 100,170 | 13,158 | 13,158 | 136,195 |

Tasks 1–4, we generate QA pairs from report text and multi-label annotations using task-specific templates, with questions scored and filtered via a rigorous LLM+human verification pipeline. For Tasks 5 and 6, we design longitudinal QA pairs by leveraging multi-phase labels to capture disease progression or resolution across time. Notably, 3D-RAD is the first 3D Med-VQA benchmark to explicitly support multi-task and multi-temporal reasoning grounded in real-world clinical scenarios.

*Open-Ended Questions (Task 1-3):* To construct data for open-ended tasks, we extract the Finding and Impression fields from the CT-RATE clinic report text. For each task, we design a corresponding prompt. Figure 4 presents a concise example of the prompt, with the comprehensive prompt detailed in Figure 11, Figure 12, and Figure 13. **For Tasks 1–2**, we construct QA pairs by injecting image-associated clinical report text into carefully designed prompts and feeding them into GPT-4o-mini[39]. **For Task 3**, we adopt a two-stage pipeline. In the first stage, we extract sentences containing quantitative measurements (e.g., "5 mm", "3×5 mm") from the clinical reports. In the second stage, these extracted sentences are inserted into prompts and input to GPT-4o-mini to generate QA pairs. To diversify question forms and reduce repetitive patterns, we adopt the "6W" framework [40, 41] and, following expert validation, select three clinically common starters—"what," "where," and "which"—for our templates. For each report, each starter is used twice to produce a list of six that is then shuffled and assigned, ensuring diversity across starters, image content, and textual cues; this strategy promotes uniqueness and balanced distribution. To mitigate redundancy, we truncate high-frequency questions or answers by retaining only the first 10 instances when an item appears more than 10 times, preserving essential semantic and contextual variety while better reflecting common clinical inquiries than extreme deduplication. We also conduct entity coverage analysis using named entity recognition to assess the distribution of anatomical entities in questions; summary statistics are reported in Figure 5, with task-wise word clouds in Figure 29 further demonstrating organ and expression diversity. To ensure the answers remain concise and aligned with the QA format—rather than drifting toward explanatory responses—we constrain generated answers to approximately five words. After the initial generation, all QA pairs undergo a post-hoc quality control process, including automatic scoring and manual sampling-based validation, resulting in a curated high-quality dataset.

*Closed-Ended Questions (Task 4-6):* We construct closed-ended question-answer pairs by selecting samples with multi-label abnormality annotations from the CT-RATE dataset. **For Task 4**, we design 18 binary classification prompts, each aligned with one of the predefined abnormality labels. To promote question diversity, the prompt pool is shuffled and assigned sequentially across labels. Answers are directly obtained from the annotations, with "0" mapped to No and "1" to Yes. **For Task 5**, we focus on patients with two or more scans. For each diagnostic label, we analyze the historical labels from the previous n–1 scans and classify the current status into one of four temporal categories. **Task 6** adopts the same answer construction strategy as Task 5, deriving responses based on findings from the previous n–1 scans. For question generation, we design three modular prompt components: *(1) temporal label templates*, *(2) interpretation templates*, and *(3) base question prompts*. These components are independently shuffled and concatenated to promote diversity. Temporal labels are generated by mapping historical scan results (1 for presence, 0 for absence) and embedded into the question. Full template details are provided in Figure 14, Figure 15, and Figure 16.

*Data Validation:* To ensure the quality of 3D-RAD, we develop a semi-automated scoring and filtering pipeline. Each generated QA pair is evaluated using a GPT-based framework across five dimensions—Visual Verifiability, Specificity & Clarity, Answer Appropriateness, Q-A Alignment,

Table 3: Finetuned Results of M3D-RAD (Llama2-7B) and M3D-RAD (Phi3-4B) on All Tasks. The best performance is highlighted in **bold**, while the second-best is underlined. Red numbers denote the performance gain over the base model after fine-tuning.

| Task | Metric | M3D (Llama2-7B) | M3D-RAD (Llama2-7B) | M3D (Phi3-4B) | M3D-RAD (Phi3-4B) |
|---|---|---|---|---|---|
| Anomaly Detection | BLEU | 9.10 | 25.25↑+16.15 | 15.06 | **33.28**↑+18.22 |
| | Rouge | 18.64 | 33.76↑+15.12 | 23.19 | **42.45**↑+19.26 |
| | BERTScore | 86.07 | 89.16↑+3.09 | 87.11 | **90.72**↑+3.61 |
| Image Observation | BLEU | 10.69 | 31.28↑+20.59 | 16.31 | **39.66**↑+23.35 |
| | Rouge | 20.82 | 39.12↑+18.30 | 23.19 | **50.52**↑+27.33 |
| | BERTScore | 86.61 | 90.00↑+3.39 | 86.92 | **92.19**↑+5.27 |
| Medical Computation | BLEU | 15.95 | 30.54↑+14.59 | 2.55 | **33.52**↑+30.97 |
| | Rouge | 23.24 | 36.06↑+12.82 | 5.63 | **36.46**↑+30.83 |
| | BERTScore | 91.50 | 94.65↑+3.15 | 85.74 | **94.86**↑+9.12 |
| Existence Detection | Accuracy | 18.00 | 81.09↑+63.09 | 40.25 | **82.43**↑+42.18 |
| Static Temporal Diagnosis | Accuracy | 25.47 | **51.20**↑+25.73 | 25.40 | 49.30↑+23.90 |
| Longitudinal Temporal Diagnosis | Accuracy | 24.17 | **74.78**↑+50.61 | 24.31 | 74.77↑+50.46 |

and Linguistic Quality—rated on a 1–5 scale. Pairs scoring below 3 in any dimension or with an average score below 3 are discarded. The remaining QA pairs are ranked by average score, and to encourage diversity, only the top 10 instances are retained for each unique question or answer. High-quality samples are then selected top-down to construct the final dataset (see Figure 4). To validate the reliability of our scoring model (GPT-4o-mini), we sample 600 QA pairs and compare its ratings with those of DeepSeek-R1 [42], LLaMA3.3-70B, and LLaMA3-8B [43] using a shared evaluation rubric. As shown in Figure 7 and Figure 8, GPT-4o-mini exhibits the highest agreement with other models on high-quality QA pairs (measured by Top-100 Overlap and NDCG@100), despite minor variation on lower-scoring samples due to scoring conservativeness. Since only high-scoring QA pairs are retained, GPT-4o-mini is adopted as the primary scoring model.

To further ensure factual alignment, we conduct human validation on the same 600 QA samples using the original radiology reports. The overall agreement rate is 91.17% (547/600), which rises to 96.17% (577/600) after excluding samples with any score below 3 or an average score below 3 across five key dimensions—confirming that most misaligned or hallucinated outputs were effectively filtered out. Final QA counts for both the benchmark and training sets, along with dataset comparisons, are summarized in Table 2.

## 3.3 Data Statistics

3D-RAD consists of two subsets: a high-quality benchmark set and a large-scale training set, 3D-RAD-T. Based on the 1,304 validation cases and 20,000 training cases in the original CT-RATE dataset, we initially generate over 400k QA pairs using our task-specific templates. To ensure quality and reduce redundancy, we perform automatic deduplication, scoring, and filtering. This results in 33,910 QA pairs for the benchmark set and 136,195 for the final training set. A detailed breakdown is provided in Table 2. To assess the linguistic and thematic diversity of 3D-RAD, we conduct a word cloud analysis across tasks (in Figure 29) [44]. The question distributions vary notably by tasks, reflecting distinct focuses while maintaining coherence within each task's clinical objective. This demonstrates the richness and task-alignment of our constructed QA corpus. To analyze the biomedical terminology covered in our benchmark dataset, we utilize the en_ner_bionlp13cg_md model[45] to extract named entities belonging to the categories PATHOLOGICAL_FORMATION, TISSUE, ORGANISM_SUBDIVISION, and ORGAN. Based on the extracted entities, we generate a unified word cloud to visualize the vocabulary distribution. Figure 5 demonstrates that our dataset encompasses a wide range of disease-related and biomedical terms, highlighting its linguistic diversity and domain coverage.

Table 4: Average Zero-Shot Performance for each VLMs on Tasks 1-6. The best performance is highlighted in **bold**, while the second-best is underlined.

| Task | Metric | RadFM | M3D (Llama2-7B) | M3D (Phi3-4B) | OmniV (Qwen2.5-1.5B) |
|---|---|---|---|---|---|
| Anomaly Detection | BLEU | 11.00 | 9.10 | **15.06** | 13.47 |
| | Rouge | 17.62 | 18.64 | 23.19 | **25.72** |
| | BERTScore | 86.76 | 86.07 | 87.11 | **88.21** |
| Image Observation | BLEU | 13.48 | 10.69 | 16.31 | **16.42** |
| | Rouge | 19.14 | 20.82 | 23.19 | **26.69** |
| | BERTScore | 87.16 | 86.61 | 86.92 | **88.29** |
| Medical Computation | BLEU | 3.34 | **15.95** | 2.55 | 2.52 |
| | Rouge | 6.62 | **23.24** | 5.63 | 7.88 |
| | BERTScore | 86.85 | **91.50** | 85.74 | 85.66 |
| Existence Detection | Accuracy | 29.20 | 18.00 | **40.25** | 28.66 |
| Static Temporal Diagnosis | Accuracy | **44.11** | 25.47 | 25.40 | 22.96 |
| Longitudinal Temporal Diagnosis | Accuracy | **42.99** | 24.17 | 24.31 | 24.23 |

# 4 Experiment

## 4.1 Experimental Setup

We evaluate several 3D-capable Med-VLMs with parameter sizes of 1.5B, 4B, 7B, and 14B, covering a range of model capacities. All models are sourced from their official Hugging Face repositories. All experiments are run on NVIDIA H200 (141 GB) and RTX 3090 (24 GB) GPUs. Experiments are conducted under two settings:

(1) **Zero-shot**, where models receive only task prompts without in-context examples, is used to evaluate generalization ability. To ensure a fair and comprehensive comparison, we benchmark several state-of-the-art medical vision-language models (VLMs) that support 3D image inputs. For each model, we adopt its latest and best-performing publicly available checkpoint. Specifically, we evaluate four recent and strong VLMs in the zero-shot setting: RadFM [31] (MedLLaMA2-13B), M3D [16] (LLaMA2-7B), M3D (Phi-3 4B), and OmniV [46] (Qwen2.5-1.5B). (2) **Fine-tuning**, where models are trained on the 3D-RAD training set to form our M3D-RAD variants, is used to evaluate the benefit of supervised adaptation. In this setting, we select two recent models with open-source training code—M3D (LLaMA2-7B) and M3D (Phi-3 4B)—and fine-tune them on 1%, 10%, and 100% of the training data.

All models are evaluated across 6 tasks. We report full fine-tuning results across all tasks in the main text, along with average performance of zero-shot. Detailed results for all subtasks are provided in the Appendix. We use macro-averaged accuracy (acc) for closed-ended tasks. For open-ended tasks, we report BLEU[47], ROUGE[48] and BERTScore[49], which respectively focus on precision, recall, linguistic variation, and semantic similarity. This diverse set of metrics allows us to evaluate model outputs from multiple perspectives beyond surface-level matching.

## 4.2 Experiment Reults

**M3D-RAD Model.** To assess the utility of 3D-RAD, we fine-tune two M3D model variants with different parameter scales, thereby constructing the M3D-RAD models. Table 3 reports the average performance across all tasks before and after fine-tuning (see Figure 18, Figure 19, Figure 20, Figure 21, and Figure 22 for detailed results). As shown, both small and large variants of M3D-RAD benefit consistently from fine-tuning across all tasks, confirming the utility of our dataset. Notably, for the newly introduced Task 5 and Task 6—both involving multi-phase reasoning—the models initially performed poorly, with accuracies around 20%. After fine-tuning with our dataset, their accuracies increased significantly, reaching over 70%, highlighting the large room for improvement in current models when dealing with temporally structured tasks. Nevertheless, despite this substantial gain, the

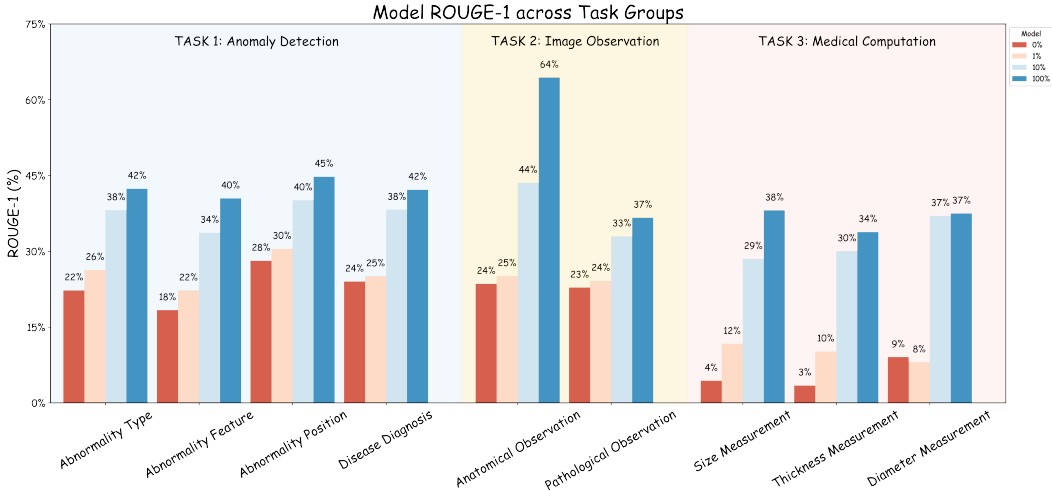

Figure 6: Finetuned Results of Models on Task 1–3. Background shading denotes each task, while bar colors indicate different proportions of the training data used for fine-tuning.

performance on these new tasks remains lower than that on Task 5, a traditional single-phase task, suggesting that while temporal cues help, they do not fully close the gap in reasoning complexity.

**Scaling with Varying Training Set Sizes.** To further investigate the impact of dataset scale on model performance, we randomly sampled 1% and 10% of the training data per task and fine-tuned M3D accordingly. The results are presented in Figure 6. We observe a consistent performance gain across all tasks as the training size increases, demonstrating the data efficiency of our benchmark. (See Appendix for subtask-level breakdown.) Moreover, we observe that current models exhibit high variance in performance on multi-phase tasks such as Task 5 and Task 6, and fail to show consistent improvement patterns with either limited or abundant supervision. This instability suggests that existing architectures may lack robust inductive bias for temporal reasoning, and that scaling data alone may be insufficient to yield reliable gains in such settings.

**Zero-Shot Evaluation on Existing Models.** We conducted zero-shot evaluation of several state-of-the-art 3D medical vision-language models on our benchmark to assess their generalization capabilities in Table 4 (see Appendix for subtask results). On conventional single-phase tasks (Task 1, Task 2, and Task 4), M3D-4B and OmniV exhibited the strongest performance. Interestingly, M3D-7B achieved superior results on Task 3, which requires numerical reasoning, suggesting stronger arithmetic capabilities. In contrast, RadFM achieved the best results on our newly proposed tasks, indicating better generalization to temporally structured QA formats. These results suggest that current models exhibit task-specific strengths, but none demonstrate universally strong performance across all clinical tasks.This task-specific variation underscores the need for more comprehensive benchmarks that span diverse clinical reasoning types—including computation, visualization, and multi-phase inference—as covered in our dataset. The complete experimental results on all subtasks are illustrated in Figure 23, Figure 24, Figure 25, Figure 26, and Figure 27.

## 5 Conclusion

In this work, we present a rigorous pipeline for constructing, filtering, and scoring high-quality QA data, resulting in 3D-RAD—a large-scale 3D medical visual question answering dataset. 3D-RAD includes both a training set and a carefully curated benchmark set, covering six task types that reflect realistic clinical scenarios. Notably, it introduces novel multi-temporal tasks that require models to assess a patient's current condition based on disease progression across multiple timepoints, closely aligning with real-world diagnostic workflows. To our knowledge, 3D-RAD is the first comprehensive Med-VQA dataset to jointly support large-scale, multi-task, and multi-temporal reasoning in 3D medical imaging. The benchmark enables robust evaluation across diverse challenges, while fine-tuning existing models on 3D-RAD yields substantial performance gains—demonstrating its potential to advance the field and establish a solid foundation for 3D medical visual understanding.

# 6 Acknowledgments

This work is supported by the National Key R&D Program of China (Grant No. 2024YFC3308304), the "Pioneer" and "Leading Goose" R&D Program of Zhejiang (Grant no. 2025C01128), the National Natural Science Foundation of China (Grant No. 62476241), the Natural Science Foundation of Zhejiang Province, China (Grant No. LZ23F020008).

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

# Contents of Technical Appendices and Supplementary Material

    **Code:** https://github.com/Tang-xiaoxiao/3D-RAD

    **Dataset:** https://huggingface.co/datasets/Tang-xiaoxiao/3D-RAD

## A    Limitations and Future Work.

Our Longitudinal Temporal Diagnosis task provides sequence information primarily from a diagnostic label perspective, which captures only one aspect of temporal evolution. However, richer temporal cues—such as spatial and morphological changes observable across full multi-phase 3D scans—remain underutilized. Current model architectures also do not support joint input of multiple 3D volumes across time, limiting comprehensive temporal reasoning. Furthermore, we have not yet introduced open-ended question formats for this task, which could enable deeper and more diverse clinical insights. In future work, we plan to incorporate full-sequence 3D inputs and develop open-ended question generation strategies to better capture longitudinal progression in medical imaging.

## B    LLM Scoring Consistency Evaluation Details.

We assess the consistency of scoring across four large language models (LLMs). The left panel in Figure 7, and Figure 8 shows the average pairwise agreement scores for each model relative to others, while the right panel presents their rankings based on overall agreement. Notably, GPT-4o-mini achieves the highest consistency, aligning closely with other models, particularly in the high-score range. Based on this result, we adopt GPT-4o as the default evaluator in our scoring pipeline.

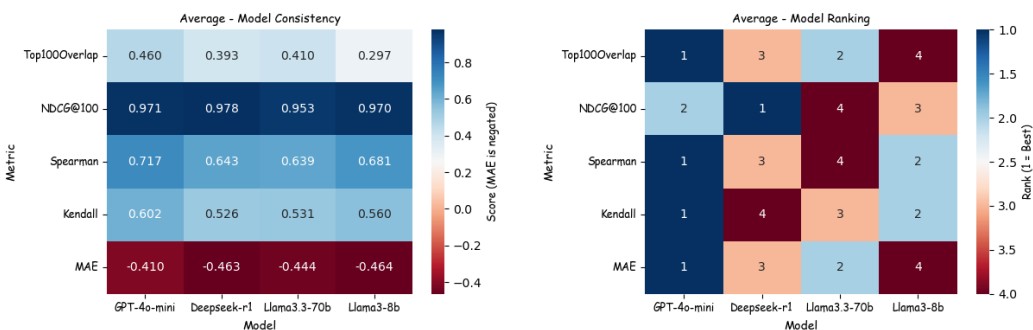

Figure 7: Consistency Heatmap          Figure 8: Ranking Heatmap

### B.1    Model Selection

To comprehensively evaluate the consistency of scoring across different large language models (LLMs), we select a diverse set of state-of-the-art models varying in architecture, scale, and training methodology:

**GPT-4o-mini**: A compact version of OpenAI's GPT-4o, optimized for fast inference with minimal performance trade-off.

**DeepSeek-r1**: A high-performing open-source LLM tailored for multilingual and multi-domain tasks, emphasizing reliability across medical and scientific content.

**LLaMA3-70B**: Meta's latest flagship 70B parameter model trained on a large corpus with improved instruction-following and reasoning capabilities.

**LLaMA3-8B**: A smaller variant of LLaMA3 designed for cost-efficient deployment while preserving alignment characteristics.

### B.2    Evaluation Metrics

To quantify inter-model scoring consistency and ranking agreement, we adopt five metrics widely used in information retrieval and statistical correlation analysis:

**Top100Overleap**: The number of overlapping QA pairs in the top-100 ranked results between model pairs, measuring agreement on high-quality samples.

**NDCG@100** (Normalized Discounted Cumulative Gain): Captures both the ranking position and relevance of QA samples up to the top 100, reflecting graded relevance alignment.

**Spearman's Rank Correlation Coefficient** ($\rho$): Measures the monotonic relationship between two sets of rankings, invariant to scale.

**Kendall's Tau** ($\tau$): Evaluates pairwise ranking agreement, more robust to small ranking perturbations than Spearman's $\rho$.

**MAE** (Mean Absolute Error): Computes the average absolute difference in numeric scores between model pairs, capturing score-level discrepancy.

## C   Evaluation Procedure.

To assess scoring consistency across models, we sample a total of 600 QA pairs from the dataset, proportionally drawn from each of the six tasks. All four models are instructed to rate these samples independently using the same standardized scoring template to ensure fairness. Each model provides five-dimensional scores per QA pair, from which we compute the average score and generate a ranked list of the 600 samples.

We then perform pairwise comparisons across all models using the metrics defined in Section B.2 For each model, we compute its average metric value against the other three models. Figure 7 illustrates the average metric values of each model relative to the others. Figure 8 shows the ranking of each model based on these averaged metrics.

Based on this analysis, we choose the **GPT-4o-mini** model for final QA scoring. It consistently demonstrates the highest agreement with the other large-scale models in terms of ranking and high-score overlap. Since our final QA set is selected based on high-scoring samples, we argue that using a model with high consistency ensures that selected QAs would also be rated highly by other LLMs, thereby preserving the objectivity and robustness of our scoring pipeline.

## D   Human Annotation Protocol.

### D.1   Annotator Background

We recruited eight graduate students with medical domain knowledge to perform manual quality assessments. Each annotator evaluated 75 QA pairs, with an average annotation time of 1.5–2 hours. To ensure consistency with the automated evaluation, all annotations followed the same scoring rubric used by LLMs. In addition, to mitigate subjectivity and provide clearer guidance, two detailed scoring examples were supplied to each annotator.

### D.2   Annotation Procedure

Annotators were provided with a comprehensive scoring manual, two reference-scored examples, and a set of sampled QA pairs along with their corresponding clinical reports (see Figure 10). Each QA pair was scored across five dimensions on a 0–5 scale: **Visual Verifiability**; **Specificity & Clarity**; **Answer Appropriateness**; **Q-A Alignment**; **Linguistic Quality**.

In addition, annotators marked whether the QA pair was consistent with the original clinical report (**binary consistency label: 1 for consistent, 0 for inconsistent**), reflecting the factual accuracy and absence of hallucination.

To ensure reliability, we first identified 53 QA pairs flagged as inconsistent (consistency score = 0). We then computed the average of the five-dimensional scores for each flagged pair and excluded any QA pair that (i) had an average score below 3 or (ii) received a score below 3 in any individual dimension. This process resulted in the retention of only 23 low-quality samples. Consequently, the final dataset achieved a high factual accuracy rate of **96.17%**.

Given the large scale of our dataset (170K QA pairs), we believe our sampling and validation protocol offers a cost-effective yet robust quality assurance mechanism, ensuring the reliability of the benchmark under practical constraints.

Table 5: Results across General VLMs on 3D-RAD Benchmark.

| Model | Task1 (ROUGE1 / BERTScore) | Task2 (ROUGE1 / BERTScore) | Task3 (ROUGE1 / BERTScore) | Task4 (ACC) | Task5 (ACC) | Task6 (ACC) |
|---|---|---|---|---|---|---|
| llava-onevision(7b) | 22.34 / 86.24 | 26.30 / 86.79 | 5.93 / 91.81 | 29.67 | 6.86 | 6.86 |
| qwen2.5-vl(3B) | 24.40 / 84.96 | 22.37 / 83.77 | 18.21 / 92.31 | 22.75 | 24.09 | 24.09 |
| qwen2.5-vl(32B) | 21.77 / 87.66 | 20.30 / 87.64 | 11.46 / 90.73 | 58.60 | 28.62 | 28.36 |
| gemma-3(4b) | 22.91 / 87.44 | 22.50 / 87.41 | 8.56 / 90.50 | 21.47 | 10.75 | 10.74 |
| gemma-3(27b) | 26.21 / 88.15 | 30.43 / 88.89 | 10.55 / 90.54 | 28.57 | 24.09 | 24.09 |
| gemini-2.0-flash | 24.70 / 88.93 | 27.93 / 89.29 | 0.53 / 84.36 | 40.42 | 23.08 | 21.33 |
| intern-vl-3(8B) | 28.44 / 88.83 | 34.31 / 89.63 | 10.65 / 91.64 | 55.30 | 26.63 | 26.61 |
| gpt-4.1-nano | 10.99 / 85.00 | 11.41 / 85.32 | 4.31 / 83.63 | 37.95 | 19.95 | 25.11 |
| gpt-4.1 | 15.08 / 86.03 | 16.55 / 86.50 | 2.76 / 82.82 | 61.98 | 34.08 | 68.05 |
| Ours-Finetuned (Llama2-7B) | 33.76 / 89.16 | 39.12 / 90.00 | 36.06 / 94.65 | 81.09 | 51.20 | 74.78 |
| Ours-Finetuned (Phi3-4B) | 42.45 / 90.72 | 50.52 / 92.19 | 36.46 / 94.86 | 82.43 | 49.30 | 74.77 |

Figure 9: An example of QA Construction

# E  Prompts and Templates.

## E.1  Question Construction Prompts and Templates.

Figure 11, Figure 12, Figure 13, Figure 14, Figure 15, and Figure 16 illustrate the complete set of carefully designed templates used for question construction across different tasks. Each template is tailored to the unique characteristics and objectives of the corresponding task, enabling accurate and diverse generation of clinically meaningful question-answer pairs.

## E.2  LLM-Based Scoring Prompt.

Figure 17 illustrates the full prompt template used to evaluate each QA pair across five key dimensions. This prompt guides large language models (LLMs) to assign quality scores, ensuring consistency and comprehensiveness in the evaluation process.

# F  Original Dataset.

Medical datasets are inherently limited and highly valuable. As illustrated in Figure 9, we present the core components of the original dataset and how we systematically leveraged this information to construct our multi-task, information-rich, high-quality VQA benchmark.

On the left, we show a representative 3D CT volume in `.nii.gz` format. The central panel contains two key types of information: **clinical text** and structured **labels**. We primarily utilized the *Findings* and *Impression* sections of the clinical text for constructing open-ended VQA pairs (Tasks 1–3). The label annotations include binary indicators for the presence of 18 diagnostic categories per scan; in cases with multiple follow-up scans for the same patient, longitudinal label comparisons are available (shown in four-scan progression examples).

Based on these sources:

- **Tasks 1–3** use clinical text and image data to generate open-ended QA pairs focusing on anomalies, anatomical structures, and measurements.
- **Task 4** uses the image-label mapping to create closed QA pairs (Yes/No) for 18 binary categories.

- **Tasks 5–6** introduce temporal reasoning. After excluding the "medical material" label (e.g., stents, which are not typically lesions), we perform longitudinal comparisons across scans on the remaining 17 labels. Each label is categorized based on its temporal evolution, with corresponding clinical implications, as follows:
  - **Refractory Lesion**: Previously 1 or fluctuating, now 1. Indicates a persistent or recurrent abnormality requiring close monitoring or intensified treatment.
  - **Resolved Lesion**: Previously 1 or fluctuating, now 0. Suggests effective resolution of the lesion, though continued surveillance may be needed to prevent recurrence.
  - **New Lesion**: Previously 0, now 1. Reflects newly emerged pathology, often signaling potential disease progression or relapse, thus necessitating timely clinical attention.
  - **No Abnormality**: Consistently 0. Denotes stable absence of pathology, typically indicating a favorable prognosis with low clinical concern.

To ensure data consistency, we exclude image variants produced by post-processing techniques (e.g., denoising) and only retain raw, unprocessed scans.

## G   QA Construction Case Visualization.

Figure 9 illustrates an example of how various task-specific information is extracted from a single radiology report. This case highlights the process of transforming complex clinical descriptions into structured QA pairs across different tasks, demonstrating the richness and multi-faceted nature of the original medical content.

In Figure 3, we demonstrate representative QA construction processes across the six tasks. This illustration highlights the key distinctions among the six diagnostic tasks in terms of input modality (e.g., text, image, label sequences), question format (open vs. closed), and reasoning requirements (e.g., spatial, numerical, temporal). By contrasting these examples side-by-side, we emphasize the diverse cognitive challenges posed by each task, which collectively test different dimensions of medical VQA capabilities.

- **Task 1:** Focused on identifying and localizing abnormalities in the image.
- **Task 2** extends beyond abnormality detection by including anatomical and structural queries that may involve normal findings, such as "Where is the cardiac pacemaker catheter terminating?", distinguishing it from Task 1 which focuses exclusively on abnormal observations.
- **Task 3:** Targets clinically relevant numerical values (e.g., diameters).
- **Task 4:** Binary classification of label existence (<Choices_list>: Yes/No) per diagnostic category.
- **Task 5:** Requires inference of current lesion status based solely on the current image,, with a four-class <Choices_list>, without longitudinal context.
- **Task 6:** Involves multi-phase diagnosis using both image features and longitudinal label progression, with a four-class <Choices_list>: Refractory Lesion (persistent or recurrent, now present); Resolved Lesion (previously present or recurrent, now absent); New Lesion (absent previously, now present); No Abnormality (always absent).

This detailed visualization supports our data construction strategy and highlights the complexity and diversity of clinical VQA scenarios tackled in our benchmark.

## H   Detailed Analysis of Experimental Results.

In the main text, we present a summary of the overall experimental results (Table 3 and Table 4). Here, we provide a more detailed analysis of these results.

### H.1   Finetuned Results of M3D-4B Models.

Table 6 presents a comprehensive comparison of the finetuned results for the 4B model series across Tasks 1 to 6. The results demonstrate consistent performance improvements after domain-specific finetuning, particularly on temporally-aware tasks (Tasks 5 and 6), highlighting the critical role of tailored 3D medical data in enhancing multi-task and multi-temporal reasoning capabilities of vision-language models.

Table 6: Finetuned Results of 4B Series on Tasks 1–6

| Task | Metric | Zero-shot | 1% | 10% | 100% |
|---|---|---|---|---|---|
| Task1: Anomaly Detection | BLEU | 15.06 | 21.10 | 30.54 | **33.28** |
| | Rouge | 23.19 | 26.03 | 37.53 | **42.45** |
| | BERTScore | 87.11 | 88.13 | 89.78 | **90.72** |
| Task2: Image Observation | BLEU | 16.31 | 20.54 | 29.35 | **39.66** |
| | Rouge | 23.19 | 24.63 | 38.25 | **50.52** |
| | BERTScore | 86.92 | 88.23 | 89.81 | **92.19** |
| Task3: Medical Computation | BLEU | 2.55 | 7.01 | 25.47 | **33.52** |
| | Rouge | 5.63 | 9.95 | 31.84 | **36.46** |
| | BERTScore | 85.74 | 85.97 | 93.06 | **94.86** |
| Task4: Existence Detection | Accuracy | 40.25 | 80.85 | 80.93 | **82.43** |
| Task5: Static Temporal Diagnosis | Accuracy | 25.40 | 41.17 | 48.11 | **49.30** |
| Task6: Longitudinal Temporal Diagnosis | Accuracy | 24.31 | 61.01 | 74.19 | **74.77** |

## H.2 Finetuned Subtask Performance.

Figure 18, Figure 19, Figure 20, Figure 21, and Figure 22 illustrate the performance comparison across subtasks after fine-tuning. The results highlight consistent improvements in most subtasks, demonstrating the effectiveness of our domain-specific training data in enhancing the model's task-specific capabilities.Notably, the most significant gains are observed in subtasks requiring temporal reasoning, suggesting that our dataset effectively supports learning nuanced clinical progressions.

## H.3 Zero-shot Subtask Performance.

Figure 23, Figure 24, Figure 25, Figure 26, and Figure 27 illustrates the performance of models under the zero-shot setting across different subtasks. The results reveal substantial variation in accuracy, highlighting that some subtasks are more tractable in the absence of fine-tuning, while others remain highly challenging. This emphasizes the varying complexity and reasoning demands posed by each subtask within our benchmark.

## H.4 Result Analysis

We evaluate several vision-language models (VLMs) on the M3D-RAD benchmark, a comprehensive suite of six carefully designed medical visual question answering (VQA) tasks. All tasks follow the same image-question-answering paradigm but differ in form and difficulty: Tasks 1–3 are open-ended generation tasks, Tasks 4–6 are closed-form classification tasks, and among them, Tasks 5 and 6 specifically evaluate temporal reasoning—Task 5 based on single-phase (static) inputs and Task 6 involving longitudinal multi-phase understanding.

**Fine-tuned Performance.** Supervision Boosts Temporal Understanding Fine-tuning significantly improves performance across all tasks, with Phi3-4B consistently outperforming LLaMA2-7B. In open-ended tasks like Anomaly Detection and Image Observation, BLEU and Rouge scores increase by 20–30 points, indicating improved alignment with radiology-style clinical descriptions.

The most dramatic gains occur in Tasks 5 and 6, which test the model's ability to infer temporal lesion status. Phi3-4B achieves 49.30% accuracy on Static Temporal Diagnosis (Task 5) and 74.77% on Longitudinal Temporal Diagnosis (Task 6), outperforming LLaMA2-7B by more than 25%–50%. These results clearly show that temporal reasoning in 3D medical data benefits greatly from task-

specific supervision, yet also reveal that existing models still struggle with this type of complex inference.

Notably, Medical Computation (Task 3) remains challenging across the board. Even with fine-tuning, generative scores (BLEU/Rouge) remain low despite high BERTScore, pointing to persistent limitations in handling structured quantitative reasoning.

**Zero-shot Performance.** Descriptive VQA Transfers Well, Temporal Reasoning Does Not In the zero-shot setting, OmniV (Qwen2.5–1.5B) achieves the strongest performance on open-ended descriptive tasks (Tasks 1–2), demonstrating strong cross-modal generalization. In contrast, RadFM outperforms all others on temporal classification tasks, with 44.11% and 42.99% on Tasks 5 and 6, respectively. This suggests that domain-specific inductive bias remains crucial for temporally grounded reasoning tasks in medical imaging.

However, even the best-performing zero-shot models exhibit significant drops in Temporal Diagnosis tasks compared to their fine-tuned counterparts—indicating that temporal lesion understanding is not emergent in current VLMs, and must be explicitly taught.

**Key Insights and Benchmark Value.** Our benchmark reveals several critical insights into current model capabilities:

- All tasks benefit from fine-tuning, but the largest improvements occur in temporally grounded VQA (Tasks 5 and 6), where supervised adaptation yields +50% accuracy gains. This indicates that temporal lesion reasoning is learnable but not captured in pretraining.

- Descriptive tasks (Tasks 1–2) transfer better to zero-shot settings, while temporal and binary classification tasks (Tasks 4–6) require explicit supervision or domain priors to perform reliably.

- Medical Computation (Task 3) consistently exposes model limitations in numerical reasoning, motivating future work on inference-aware VQA models.

- Most importantly, our benchmark is the first to systematically expose these weaknesses across diverse VQA tasks, especially in multi-phase lesion diagnosis. By explicitly separating static and longitudinal reasoning, and grounding all tasks in real 3D CT scans with clinical language, M3D-RAD offers a fine-grained diagnostic lens into the limitations of current multimodal models—and a clear path forward for future method development.

# I   Failure Case Analysis

To better understand model limitations in temporal reasoning, we provide a detailed analysis of representative failure cases in Figure 28. We focus on Tasks 5 and 6—*Static Temporal Diagnosis* and *Longitudinal Temporal Diagnosis*—and compare the performance of zero-shot and fine-tuned models.

**Case 1: Both Task 5 and Task 6 failed in zero-shot, and fine-tuning only improved Task 6.** In this case, the lesion had resolved, but both tasks were misclassified as *Refractory Lesion* by the zero-shot model. Fine-tuning improved longitudinal reasoning (Task 6), correctly identifying the lesion as *Resolved*, while Task 5 remained incorrect. This highlights the challenge of inferring temporal status from a single image without explicit sequential cues.

**Case 2: Zero-shot failed on both tasks; fine-tuning successfully corrected both.** This case demonstrates that fine-tuning effectively leverages subtle spatial features in Task 5 and temporal patterns in Task 6. The consistent improvement suggests the model benefits from domain-adapted learning to distinguish patterns like disappearance of subtle abnormalities.

**Case 3: Task 5 failed in both conditions; Task 6 succeeded without fine-tuning.** Here, the model incorrectly inferred a *No Abnormality* outcome in Task 5. Despite this, the zero-shot model correctly answered Task 6, likely due to explicit access to the label sequence. This underscores that Task 6 provides more external structure, making it more robust even without fine-tuning.

**Case 4: Task 5 failed in zero-shot but succeeded after fine-tuning; Task 6 succeeded in both.** This case illustrates the potential of fine-tuning to teach the model to recognize spatial indicators of stable lesions in a single image. The strong performance on Task 6 across both settings affirms that longitudinal cues (label sequences) serve as a stabilizing prior.

Table 7: Evaluation accuracy on Task4–Task6 under different finetuning settings.

| Finetuned Task(s) | Evaluate Task4 (ACC) | Evaluate Task5 (ACC) | Evaluate Task6 (ACC) |
|---|---|---|---|
| Only Task1 | 82.03 | 38.78 | 72.61 |
| Only Task2 | 81.88 | 48.73 | 73.55 |
| Only Task3 | 81.94 | 48.30 | 72.78 |
| Only Task4 | 81.96 | 38.97 | 73.52 |
| Zero-Shot M3D (Llama2-7B) | 18.00 | 25.47 | 24.17 |
| Zero-Shot M3D (Phi3-4B) | 40.25 | 25.40 | 24.31 |
| All Tasks (Ours) | 82.43 | 49.30 | 74.77 |

Table 8: Evaluation metrics on Task1 (BLEU / ROUGE1 / F1) under different finetuning settings.

| Finetuned Task(s) | Evaluate Task1 (BLEU / ROUGE1 / F1) |
|---|---|
| Only Task5 | 31.60 / 42.01 / 90.48 |
| Only Task6 | 32.45 / 42.87 / 90.73 |
| Zero-Shot M3D (Llama2-7B) | 9.10 / 18.64 / 86.07 |
| Zero-Shot M3D (Phi3-4B) | 15.06 / 23.19 / 87.11 |
| All Tasks (Ours) | 33.28 / 42.45 / 90.72 |

**Summary.** Our analysis reveals distinct challenges for Tasks 5 and 6: Task 5 requires spatial-temporal inference from a single snapshot, which proves difficult without targeted training; in contrast, Task 6 benefits from the presence of explicit sequential labels, making it more amenable to reasoning even in zero-shot. Fine-tuning significantly improves performance, particularly for Task 5, by embedding temporal priors into image understanding.

## J   Word Cloud Visualizations across Tasks.

To illustrate the diversity of our benchmark, we present word cloud visualizations for each individual task. These visualizations highlight the varying distributions of clinical concepts across tasks, reflecting the distinct focuses and linguistic patterns inherent to each QA setting. Figure 29 shows the task-wise word clouds.

## K   Additional Evaluations on General VLMs

We also evaluate our benchmark on mainstream general-purpose vision–language models.As illustrated in Table 5, though some models perform better—especially on tasks 4–6 with more reliable metrics, none achieve strong results across all tasks. Models fine-tuned on 3D-RAD consistently outperform general models, highlighting the dataset's value in promoting domain-specific learning.

## L   Task-level Ablation Studies

We added task-level ablation studies to examine cross-task effectiveness in Table 7 and Table 8. The table below presents results where each task is fine-tuned individually and then evaluated on all other tasks. As shown, fine-tuning on a single task can still improve performance on other tasks. However, fine-tuning on the full dataset consistently yields the best results. This demonstrates that the effectiveness of our dataset stems not only from its size but also from the inclusion of high-quality domain knowledge.

Please score the following radiology visual question and answer pair, using the 5 dimensions below. Each should be scored from 1 (very poor) to 5 (excellent). Be strict, specific, and consistent in your evaluation.

Question: {question}
Answer: {answer}

Scoring Dimensions:

*1. Visual Verifiability: Can this question be answered just by looking at the image, without requiring medical knowledge, inference, or external context? Does the answer also rely on the image for validation?*
   - 5: Can be answered purely from the image (e.g., "Is there a fracture?") and the answer is clearly image-based (e.g., "Yes, there is a fracture in the left femur.")
   - 1: Requires background knowledge, inference, or external context (e.g., "What type of tumor?") and the answer similarly relies on external context rather than the image.
   - If the answer requires knowledge of clinical context or interpretation beyond the image itself, it should be scored lower.

*2. Specificity & Clarity: Is the question precise and unambiguous? Is the answer also specific and clear, without ambiguity?*
   - 5: The question and answer are both specific and clear with one unambiguous interpretation (e.g., "Which lobe is involved in the lesion?" and the answer directly states the affected lobe).
   - 1: The question or the answer is vague, ambiguous, or open to multiple interpretations (e.g., "What is abnormal?" and the answer gives an unclear response like "There is something unusual.")
   - Avoid vague terms like "a few," "several," "some," "few millimeters," or similar imprecise quantities. If these terms appear in the answer, it should be scored lower.
   - Both the question and the answer must be precise. If either is unclear, reduce the score.

*3. Answer Appropriateness: Is the answer correct, medically appropriate, specific, and directly relevant to the question? Does it match the expected format/type?*
   - 5: Accurate, specific, and directly answers the question with no irrelevant details (e.g., "There is a mass in the right upper lobe.").
   - 1: Inaccurate, vague, or only loosely related to the question (e.g., "There might be something abnormal.")
   - If the answer contains errors, vague descriptions, or misinterprets the question, it should be scored lower. The answer should directly match the content of the question.

*4. Q-A Alignment: Does the answer format/type match the question format/type? Is the answer logically aligned with the type of information requested in the question?*
   - 5: Perfect match (e.g., question asks "Where", answer gives a location; question asks "What size", answer gives a size).
   - 1: Mismatch in type or category (e.g., question asks "What", but the answer gives "When").
   - Ensure the answer matches the expected type (e.g., answering a "Where" question with a location, answering a "What" question with a description of the object or condition).

*5. Linguistic Quality: Are both the question and the answer grammatically correct, fluent, and easy to understand? Are there any issues with language clarity in either the question or the answer?*
   - 5: Both question and answer are fluent, clear, and easy to understand without grammatical issues or awkward phrasing.
   - 1: Question or answer is awkward, grammatically incorrect, or confusing.
   - If the answer contains significant linguistic errors that cause confusion or misunderstandings, it should be scored lower. Also, check for spelling and grammar.

Example 1:
VolumeName, Visual Verifiability, Specificity & Clarity, Answer Appropriateness, Q-A Alignment, Linguistic Quality
test_806_a_2.nii.gz,4,4,5,5,5
Question: Which structures are midline?
Answer: Trachea, bronchi
Clinic Text: Trachea and both main bronchi were in the midline and no obstructive pathology was observed in the lumen…
Reasons:
**Visual Verifiability (4):** "Trachea" is clearly midline and visually verifiable; "bronchi" is ambiguous without specifying main bronchi.
**Specificity & Clarity (4):** The term "bronchi" is too general and could refer to various branches, reducing clarity.
**Answer Appropriateness (5):** The answer includes anatomically relevant and reasonable structures.
**Q-A Alignment (5):** The answer directly responds to the question about midline structures.
**Linguistic Quality (5):** Fluent and grammatically correct.

Example 2:
VolumeName, Visual Verifiability, Specificity & Clarity, Answer Appropriateness, Q-A Alignment, Linguistic Quality
test_63_b_2.nii.gz,5,5,5,5,5
Question: What is the diameter of the main pulmonary artery?
Answer: 36mm
Clinic Text: …The diameter of the main pulmonary artery was 36mm, the diameter of the right pulmonary artery was 28mm, and the diameter of the left pulmonary artery was 25mm, showing dilatation…
Reasons:
**Visual Verifiability (5):** Main pulmonary artery diameter is a directly measurable anatomical feature.
**Specificity & Clarity (5):** The answer is precise ("36mm") and unambiguous.
**Answer Appropriateness (5):** Value is correctly extracted and clinically appropriate.
**Q-A Alignment (5):** The answer directly matches the question.
**Linguistic Quality (5):** Answer is concise and grammatically correct.

Figure 10: Guideline

**Task 1-2 Prompts**

**system_text** = """
   You are a medical AI visual assistant that can analyze a single CT image. You receive the medical diagnosis report. The report describes multiple abnormal lesions in the image.
   The task is to use the report information to create 6 questions and answers about the image.
   The first five questions and answers must be based strictly on the Findings text, and the sixth question and answer must be based strictly on the Impressions text.

   These questions come from the following 6 aspects:
   *1). Anatomical observation (based on Findings)*
   *2). Pathological observation (based on Findings)*
   *3). Abnormality type (based on Findings)*
   *4). Abnormality feature (based on Findings)*
   *5). Abnormality position (based on Findings)*
   *6). Abnormality or normality diagnosis (based on Impressions)*
   """

**PROMPT** = """
Clinical text: {
Findings: <Findings>
Impressions: <Impressions>
}

Please generate a set of exactly 6 clinical image-based question-answer pairs, strictly following these constraints:

- Focus the questions on image features related to 6 aspects, with the first five questions generated strictly based on "Findings", and the sixth question based strictly on "Impressions".
- Do not reference, mention, or imply the words "findings" or "impressions" in any part of the question. Questions must not use phrases like "assessment", "based on findings", or "what is noted" that imply summary or interpretation.
- Treat the content as if it comes from direct image observation, not from a text report.
- Avoid overly broad or vague questions. Ensure each question is specific, objective, visually verifiable, and based solely on image evidence.
- Questions must only be answerable by directly observing the image — not from general knowledge, not through inference, and not through assumptions.
- Do not generate a question that can be answered without the image.
- Avoid generating questions that require medical calculations such as number, size, volume, or specific coordinate locations.
- The answer must directly correspond to the question asked, based on the content of the clinical text.
- The answers and the questions must conform to correct medical knowledge. Ensure that the answer is both clinically relevant to the image and accurate.
- The answers should describe observable visual aspects of the image in concise phrases, using no more than 3 words.
- Please do not ask directly what organs or abnormalities are visible in the image, as the answers are not unique.
- Ensure that the question only has one clear, unique answer that would be consistently given by different people analyzing the same image.
- Avoid overly broad or vague questions. Ensure each question is specific, objective, and visually verifiable.
- The questions and answers should assume that the task is to be performed based on the image alone. Do not mention the report in the questions and answers.
- The generated questions should begin with <Starter> in a way that is natural and coherent, avoiding awkward or forced phrasing.
- The questions should not be overly complicated and should be easy for both AI models and doctors to answer accurately.
- The questions and answers must be strictly aligned; the answer must match the question type.
- For "Where" questions, the answer must be a specific anatomical location, not an appearance or visibility description.
- The answers should be directly accurate.

Desired format:
1). Anatomical observation (based on Findings)
Question-1: <Starter1> ...? Answer: ...\n
2). Pathological observation (based on Findings)
Question-2: <Starter2> ...? Answer: ...\n
3). Abnormality type (based on Findings)
Question-3: <Starter3> ...? Answer: ...\n
4). Abnormality feature (based on Findings)
Question-4: <Starter4> ...? Answer: ...\n
5). Abnormality position (based on Findings)
Question-5: <Starter5> ...? Answer: ...\n
6). Abnormality or normality diagnosis (based on Impressions)
Question-6: <Starter6> ...? Answer: ...\n
"""

Figure 11: Prompts of Task 1-2

**Task 3 Prompts (First Stage)**

**system_text** = """
    You are a medical AI visual assistant that can analyze a single CT image. You receive the medical diagnosis report. The report describes multiple abnormal lesions in the image.
    The task is to extract all complete sentences from the text that contain numerical values.
"""

**PROMPT** = """
    Please extract all complete sentences that contain specific numerical values from the following radiology finding report.
    - Keep the original expression of the sentences without making any modifications.
    - The extracted sentences should be applicable for creating a medical quantitative question and answer high-quality dataset from these three perspectives: 1). Size 2). Diameter 3). Thickness.
    - Quantities should be specific, estimates such as "a few" should not be included.
    - If no suitable sentences are found, no output is necessary.

    Radiology finding report:
    <Findings>

    Desired format:
    1). Sentence-1: ...\n
    2). Sentence-2: ...\n
    ...
"""

Figure 12: Prompts of Task 3 (First Stage)

## Task 3 Prompts (Second Stage)

**system_text** = """
 You are a medical AI visual assistant that can analyze a single CT image. You receive the medical diagnosis report. The report describes multiple abnormal lesions in the image.
 The task is to use the report information to choose the most related aspect and create one question and answer about the image.
 Each question must include a specific descriptor or location to uniquely identify the lesion or nodule being referred to.
 Answers must be exact and not estimations or imaginary numbers.

 The question and answer come from one of the following 3 aspects:
 *1). Size*
 *2). Diameter*
 *3). Thickness*
"""

**PROMPT** = """
 Based on the following radiology description sentence, choose the most appropriate aspect from the four options and generate one quantitative medical question and its corresponding answer.

 - Include a specific description or location in the question to uniquely identify the nodule or lesion being referred to.
 - Do not reference, mention, or imply the words "findings" or "impressions" in any part of the question. Questions must not use phrases like "assessment", "based on findings", or "what is noted" that imply summary or interpretation.
 - Treat the content as if it comes from direct image observation, not from a text report.
 - Avoid overly broad or vague questions. Ensure each question is specific, objective, visually verifiable, and based solely on image evidence.
 - Questions must only be answerable by directly observing the image — not from general knowledge, not through inference, and not through assumptions.
 - Do not generate a question that can be answered without the image.
 - Avoid generating questions that require medical calculations such as number, size, volume, or specific coordinate locations.
 - The answer must directly correspond to the question asked, based on the content of the clinical text.
 - The answers and the questions must conform to correct medical knowledge. Ensure that the answer is both clinically relevant to the image and accurate.
 - The answers should describe observable visual aspects of the image in concise phrases, using no more than 3 words.
 - Please do not ask directly what organs or abnormalities are visible in the image, as the answers are not unique.
 - Ensure that the question only has one clear, unique answer that would be consistently given by different people analyzing the same image.
 - Avoid overly broad or vague questions. Ensure each question is specific, objective, and visually verifiable.
 - The questions and answers should assume that the task is to be performed based on the image alone. Do not mention the report in the questions and answers.
 - The questions should not be overly complicated and should be easy for both AI models and doctors to answer accurately.
 - The questions and answers must be strictly aligned; the answer must match the question type.
 - For "Where" questions, the answer must be a specific anatomical location, not an appearance or visibility description.
 - The answers should be directly accurate.

 The question comes from the following 3 aspects:
 1). Size
 2). Diameter
 3). Thickness

 Radiology description sentence:
 <Sentence>

 Desired format:
 Aspect(Size, Diameter or Thickness)
 Question: ...? Answer: ...
"""

Figure 13: Prompts of Task3 (Second Stage)

question_patterns = [
  "Is there any presence of {aspect} on the CT scan?",
  "Can {aspect} be detected on the CT scan?",
  "Is {aspect} visible on the CT images?",
  "Does the CT scan suggest {aspect}?",
  "Can {aspect} be observed on the CT scan?",
  "Is there evidence of {aspect} on the CT images?",
  "Is {aspect} identifiable on the CT scan?",
  "Does this CT image show {aspect}?",
  "Is {aspect} present on the CT scan?",
  "Does the CT scan demonstrate {aspect}?",
  "Can {aspect} be clearly seen on the CT scan?",
  "Is {aspect} apparent on the CT images?",
  "Does the CT scan reveal {aspect}?",
  "Is {aspect} confirmed by the CT findings?",
  "Can signs of {aspect} be recognized on the CT scan?",
  "Does this CT image present {aspect}?",
  "Is {aspect} noted on the CT images?",
  "Does the CT scan exhibit {aspect}?"
]
  53

Figure 14: Prompts of Task4

Figure 15: Prompts of Task5

**Task 6 Prompts**

**question_aspects** = [
  "Arterial wall calcification",
  "Cardiomegaly",
  "Pericardial effusion",
  "Coronary artery wall calcification",
  "Hiatal hernia",
  "Lymphadenopathy",
  "Emphysema",
  "Atelectasis",
  "Lung nodule",
  "Lung opacity",
  "Pulmonary fibrotic sequela",
  "Pleural effusion",
  "Mosaic attenuation pattern",
  "Peribronchial thickening",
  "Consolidation",
  "Bronchiectasis",
  "Interlobular septal thickening"
]

**fluctuation_prefixes** = [
  "Based on sequence history and current CT.",
  "From multi-stage sequences and present CT.",
  "Sequence trend plus current CT defines status.",
  "History and CT decide lesion category.",
  "Stage-wise sequences + current CT = status.",
  "Past sequences and CT determine lesion type.",
  "Multi-phase sequences guide CT-based judgment.",
  "Classification uses sequences and current image.",
  "Lesion state from timeline and CT.",
  "CT reflects pattern of prior sequences.",
  "Lesion judged by sequences and scan.",
  "Sequence evolution and CT define outcome.",
  "Diagnosis combines sequence history and CT.",
  "CT confirms what sequences suggest.",
  "Lesion behavior from past to CT.",
  "CT status follows sequence progression.",
  "Final label from history and CT."
]

**sequence_prefixes** = [
  "{aspect} sequence was: {seq}.",
  "Past sequences of {aspect}: {seq}.",
  "Previous {aspect} states: {seq}.",
  "Earlier {aspect} sequences: {seq}.",
  "Historical {aspect} status: {seq}.",
  "Sequence history for {aspect}: {seq}.",
  "Prior {aspect} timeline: {seq}.",
  "Scans showed {aspect} as: {seq}.",
  "Old sequences for {aspect}: {seq}.",
  "Earlier CTs showed {aspect} as: {seq}.",
  "Before now, {aspect} showed {seq}.",
  "{aspect} had: {seq} in earlier scans.",
  "{aspect} progression: {seq}.",
  "In past, {aspect} was: {seq}.",
  "{aspect} over time: {seq}.",
  "{aspect} condition history: {seq}.",
  "Recorded sequences for {aspect}: {seq}."
]

**question_patterns** = [
  "What is the current status of {aspect} based on previous sequences?", "Based on past sequences, what is the current condition of {aspect}?",
  "How does {aspect} appear now compared to its sequence history?", "What type of lesion is {aspect} now, given its temporal sequence?",
  "From sequence history to now, what best describes {aspect}?","How has {aspect} changed from past sequences to the current CT?",
  "What does the current CT show about {aspect} based on earlier sequences?", "Given its sequence history, what is {aspect} in the current CT?",
  "How has {aspect} evolved from earlier scan sequences to now?", "What lesion category applies to {aspect} in the current CT?",
  "How is {aspect} classified now using past sequence data?", "What does the CT show for {aspect} considering its sequence history?",
  "What is the current CT assessment of {aspect} based on prior sequences?", "How has {aspect} progressed according to its temporal sequence?",
  "What does {aspect} currently represent, based on past sequences?", "Based on the temporal sequence, what is the present status of {aspect}?",
  "From prior sequences to now, what is the CT-based status of {aspect}?"
]

**label_map** = {
  "A": "Refractory Lesion (Persistent or recurrent, now present)", "B": "Resolved Lesion (Previously present or recurrent, now absent)",
  "C": "New Lesion (Absent previously, now present)","D": "No Abnormality (Always absent)"
}

Figure 16: Prompts of Task6

---

**Check**

---

**system_text** = """
    You are a medical expert specializing in radiology, particularly in chest CT imaging. Your task is to score the following radiology question and answer pairs based on specific criteria. You will evaluate each pair using the five dimensions listed below. Be strict, specific, and consistent in your evaluation, and ensure the scores reflect the quality of the content in the context of chest CT findings.
    """

**PROMPT** = """
    Please score the following radiology visual question and answer pair, using the 5 dimensions below. Each should be scored from 1 (very poor) to 5 (excellent). Be strict, specific, and consistent in your evaluation.

    Question: {question}
    Answer: {answer}

    Scoring Dimensions:

    *1. Visual Verifiability: Can this question be answered just by looking at the image, without requiring medical knowledge, inference, or external context? Does the answer also rely on the image for validation?*
        - 5: Can be answered purely from the image (e.g., "Is there a fracture?") and the answer is clearly image-based (e.g., "Yes, there is a fracture in the left femur.")
        - 1: Requires background knowledge, inference, or external context (e.g., "What type of tumor?") and the answer similarly relies on external context rather than the image.
        - If the answer requires knowledge of clinical context or interpretation beyond the image itself, it should be scored lower.

    *2. Specificity & Clarity: Is the question precise and unambiguous? Is the answer also specific and clear, without ambiguity?*
        - 5: The question and answer are both specific and clear with one unambiguous interpretation (e.g., "Which lobe is involved in the lesion?" and the answer directly states the affected lobe).
        - 1: The question or the answer is vague, ambiguous, or open to multiple interpretations (e.g., "What is abnormal?" and the answer gives an unclear response like "There is something unusual.")
        - Avoid vague terms like "a few," "several," "some," "few millimeters," or similar imprecise quantities. If these terms appear in the answer, it should be scored lower.
        - Both the question and the answer must be precise. If either is unclear, reduce the score.

    *3. Answer Appropriateness: Is the answer correct, medically appropriate, specific, and directly relevant to the question? Does it match the expected format/type?*
        - 5: Accurate, specific, and directly answers the question with no irrelevant details (e.g., "There is a mass in the right upper lobe.").
        - 1: Inaccurate, vague, or only loosely related to the question (e.g., "There might be something abnormal.")
        - If the answer contains errors, vague descriptions, or misinterprets the question, it should be scored lower. The answer should directly match the content of the question.

    *4. Q-A Alignment: Does the answer format/type match the question format/type? Is the answer logically aligned with the type of information requested in the question?*
        - 5: Perfect match (e.g., question asks "Where", answer gives a location; question asks "What size", answer gives a size).
        - 1: Mismatch in type or category (e.g., question asks "What", but the answer gives "When").
        - Ensure the answer matches the expected type (e.g., answering a "Where" question with a location, answering a "What" question with a description of the object or condition).

    *5. Linguistic Quality: Are both the question and the answer grammatically correct, fluent, and easy to understand? Are there any issues with language clarity in either the question or the answer?*
        - 5: Both question and answer are fluent, clear, and easy to understand without grammatical issues or awkward phrasing.
        - 1: Question or answer is awkward, grammatically incorrect, or confusing.
        - If the answer contains significant linguistic errors that cause confusion or misunderstandings, it should be scored lower. Also, check for spelling and grammar.

    Example response format:
    {{
     "Visual Verifiability": 3,
     "Specificity & Clarity": 4,
     "Answer Appropriateness": 5,
     "Q-A Alignment": 5,
     "Linguistic Quality": 4
    }}
    """

Figure 17: Prompts of Check

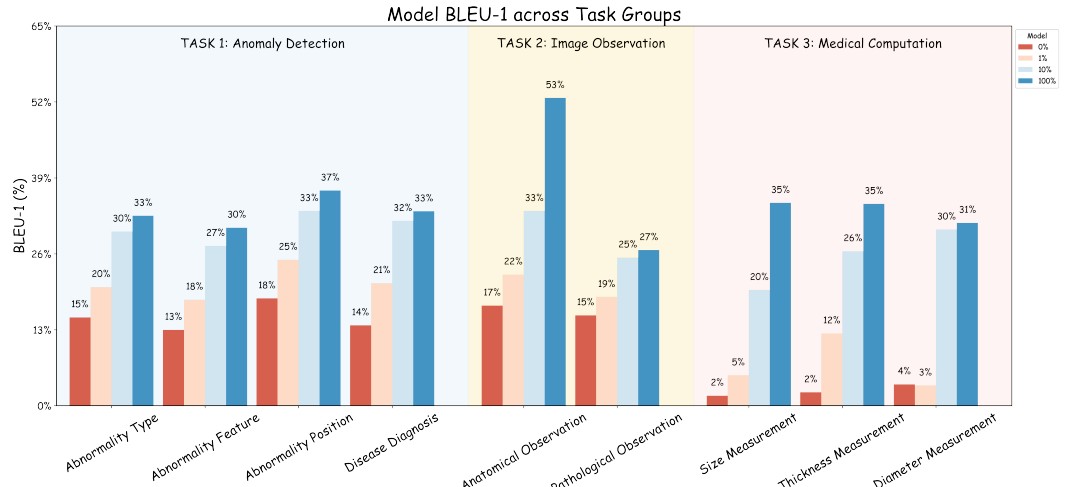

Figure 18: Finetuned BLEU Result of Task 1-3

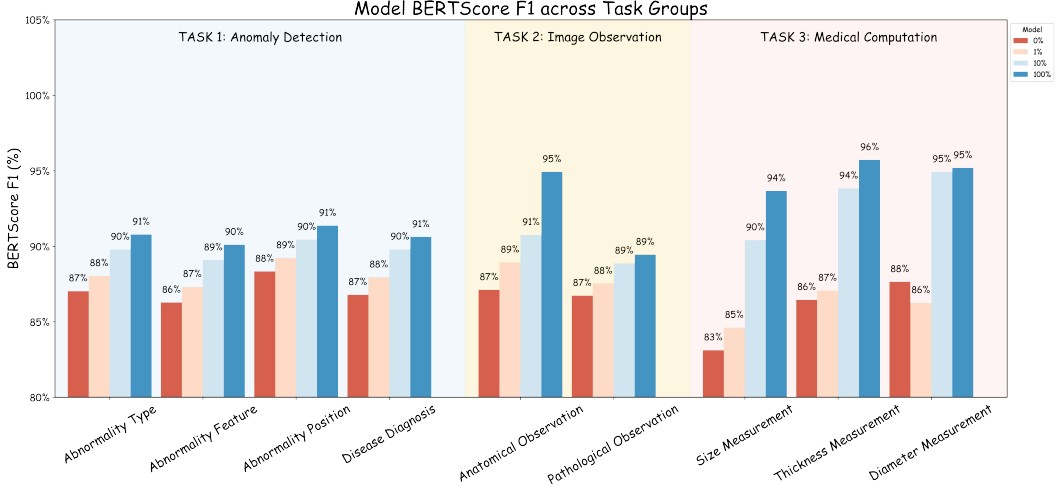

Figure 19: Finetuned F1 Result of Task 1-3

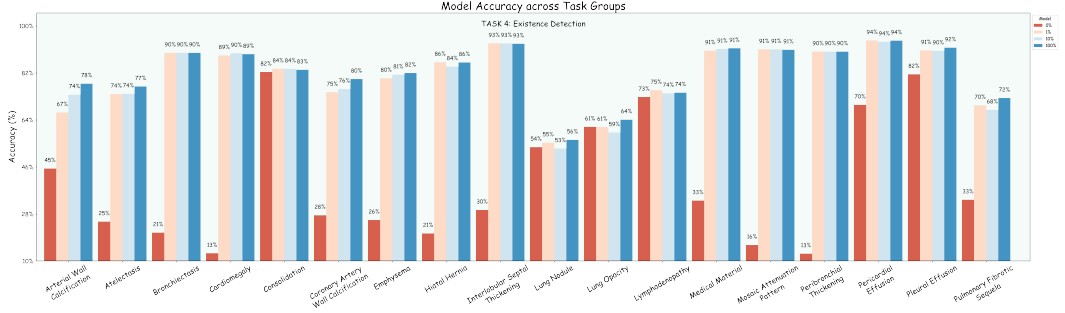

Figure 20: Finetuned results of models on Task 4

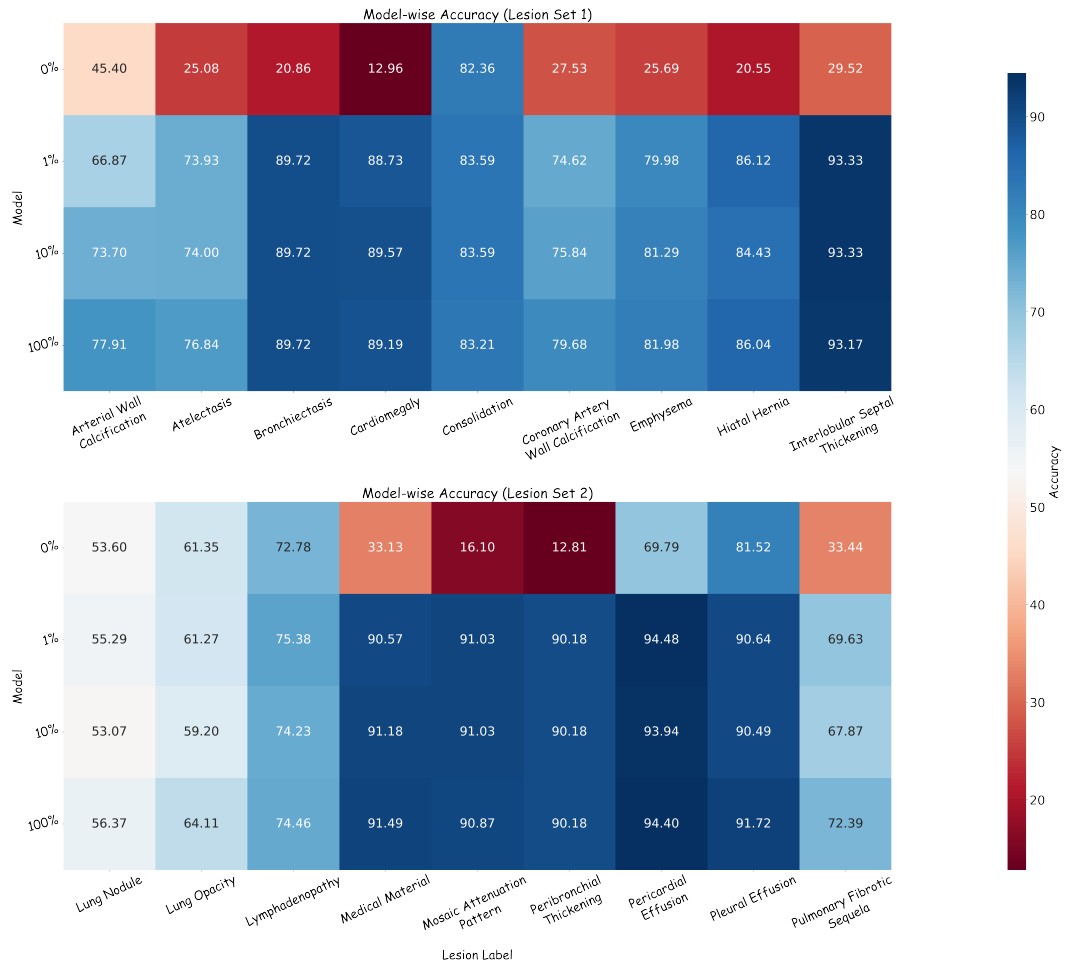

Figure 21: Finetuned Result of Task 4

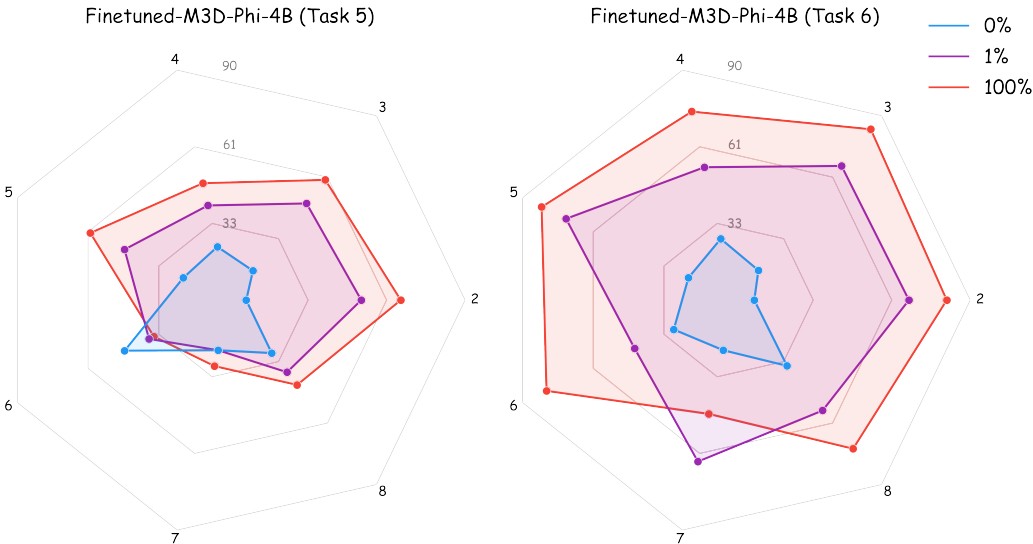

Figure 22: Finetuned Result of Task 5-6

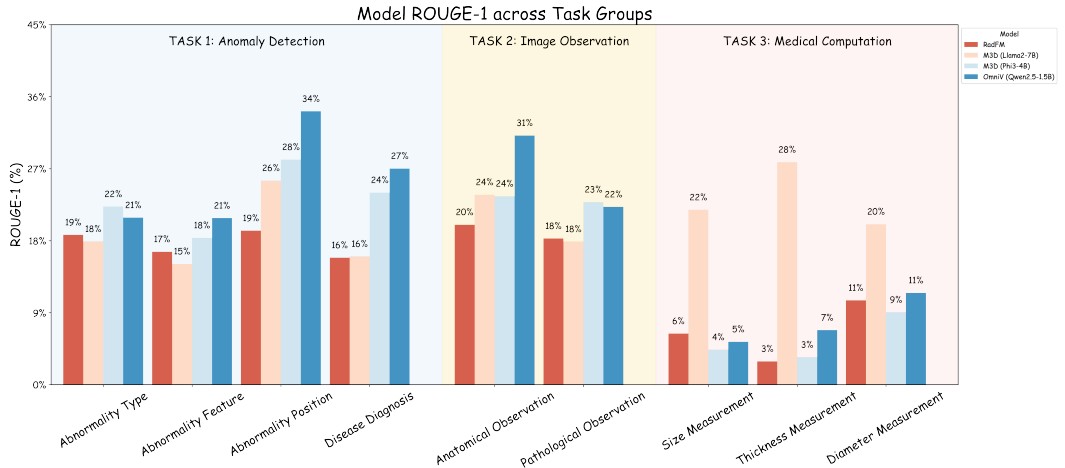

Figure 23: Zero-Shot Rouge Result of Task 1-3

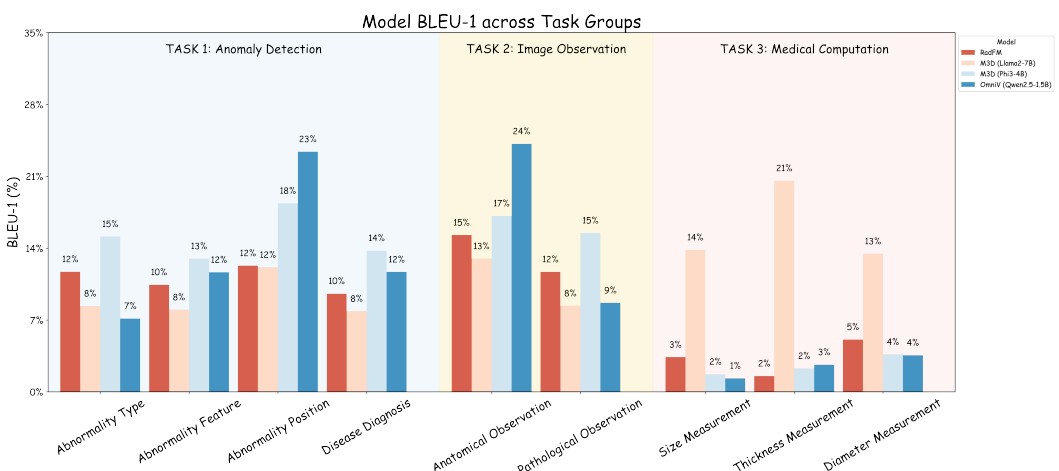

Figure 24: Zero-Shot BLEU Result of Task 1-3

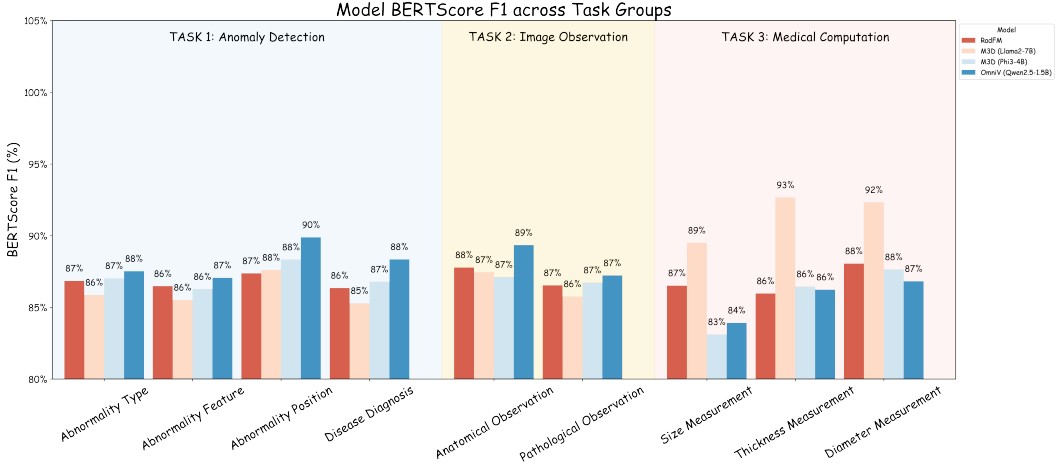

Figure 25: Zero-Shot F1 Result of Task 1-3

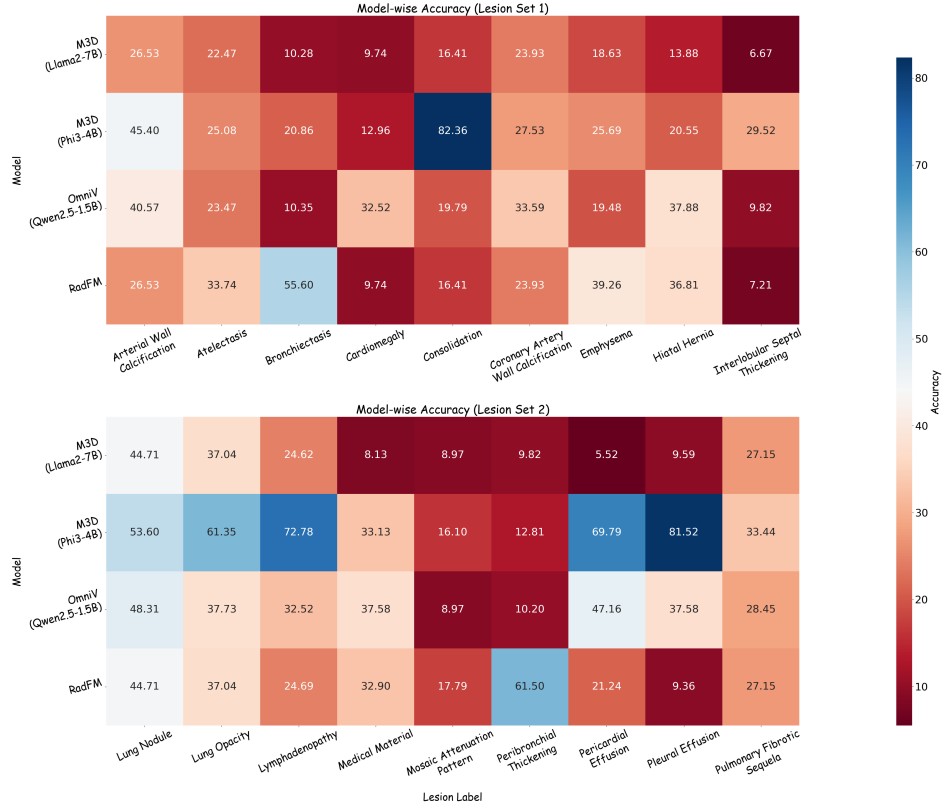

Figure 26: Zero-Shot Result of Task 4

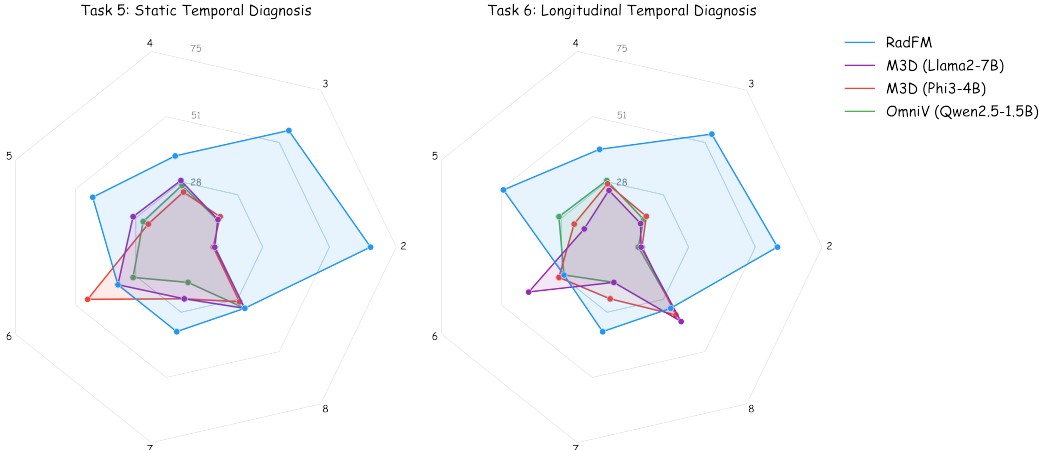

Figure 27: Zero-Shot Result of Task 5-6

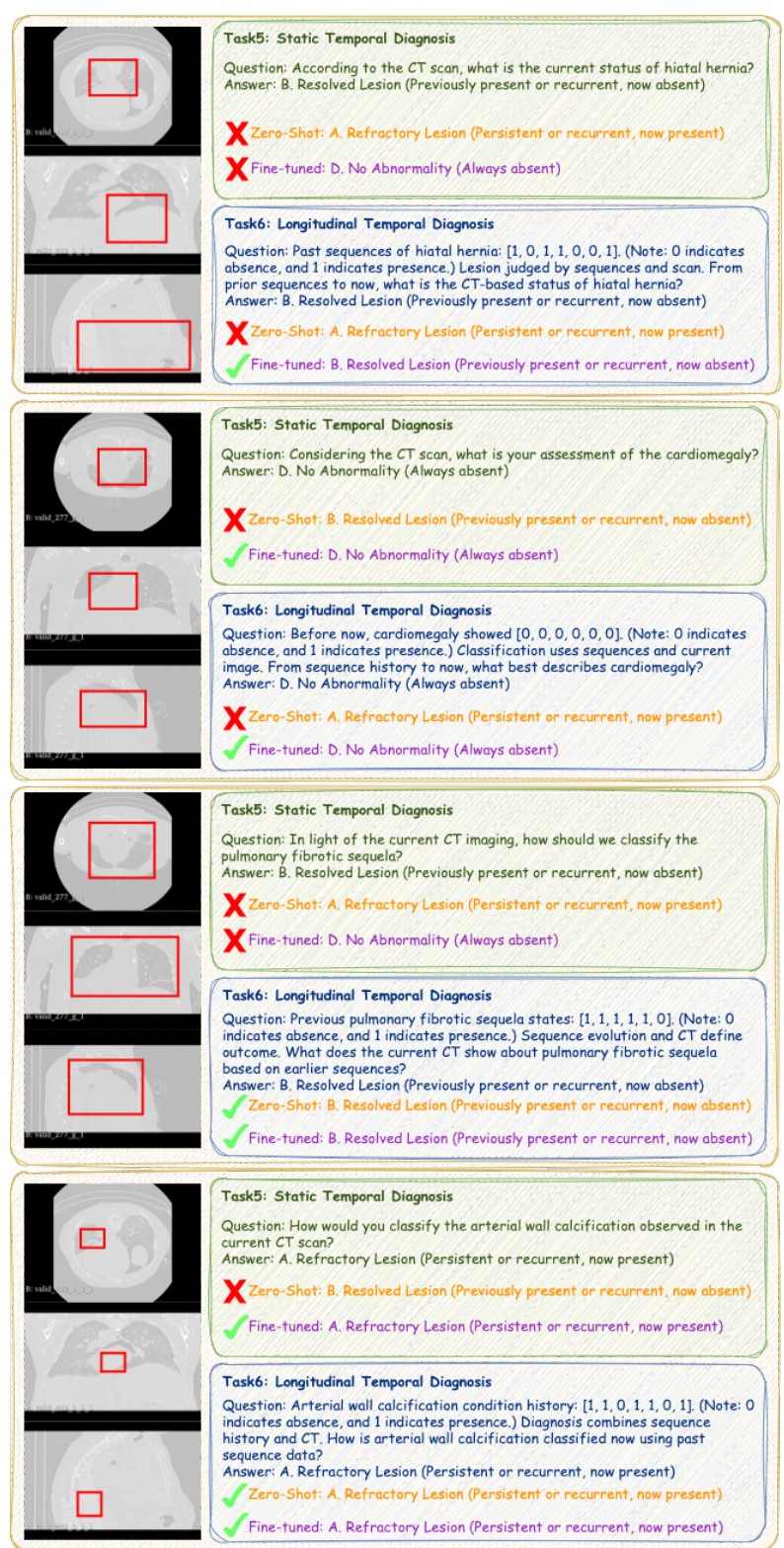

Figure 28: Failure case visualization for Tasks 5 and 6. Green checkmarks indicate correct answers; red crosses indicate failures. Each row pair shows performance of zero-shot (top) and fine-tuned (bottom) models for both task types.

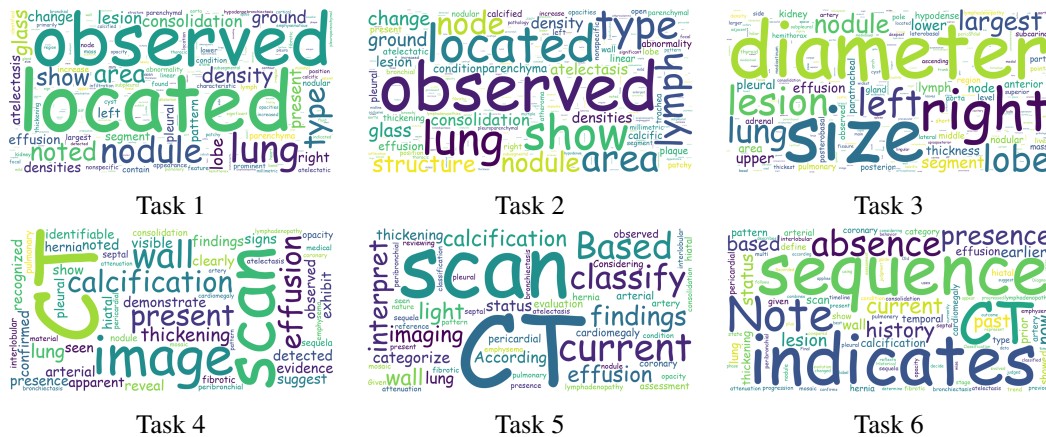

Figure 29: Wordclouds of task-specific question content across Task 1 to Task 6.

