# OpenReview forum: "3D-RAD: A Comprehensive 3D Radiology Med-VQA Dataset with Multi-Temporal Analysis and Diverse Diagnostic Tasks"
_NeurIPS.cc/2025/Datasets_and_Benchmarks_Track — NeurIPS 2025 Datasets and Benchmarks Track poster_

### Official Review · Reviewer_e2md · 2025-06-30

**Rating:** 5
**Confidence:** 3

**Summary:**

The submission introduces 3D-RAD, a large-scale dataset designed to advance 3D Medical Visual Question Answering (Med-VQA) using radiology CT scans. It encompasses six diverse VQA tasks and supports both open- and closed-ended questions, introducing complex reasoning challenges. Evaluations show limited generalization of existing vision-language models (VLMs), especially in multi-temporal tasks. The paper also releases a high-quality training set, 3D-RAD-T, with 136,195 expert-aligned samples to enhance model performance.

**Dataset Code Accessibility:**

Yes

**Dataset Code Comments:**

Yes, this study has provided dataset and code links.

**Ethical Considerations:**

No, there are no or only very minor ethics concerns

**Final Justification:**

The rebuttal has addressed all of my concerns. Therefore, I have raised my score to ‘Accept’.

**Limitations Weaknesses:**

1. The dataset is named 3D-RAD, yet it exclusively focuses on CT scans, limiting its applicability to other crucial 3D imaging modalities such as MRI and PET. The authors are encouraged to revise the dataset name to more accurately reflect its current scope.
2. The study defines six VQA tasks and constructs the dataset accordingly. Table 3 highlights the overall positive impact of the dataset on these tasks. However, no ablation studies are provided to examine cross-task effectiveness. For instance, it remains unclear whether fine-tuning on one task could improve performance on another (e.g., in zero-shot or few-shot settings).
3. The paper lacks experiments evaluating the transferability of the fine-tuned models to the M3D-VQA dataset.

**Strengths Contributions:**

1. 3D-RAD is the first comprehensive dataset to support multi-task and multi-temporal reasoning in 3D medical imaging, addressing the limitations of existing 2D-focused Med-VQA datasets like VQA-RAD, SLAKE, and PathVQA, which lack 3D support and temporal analysis.
2. The dataset encompasses six diverse VQA tasks, supporting both open- and closed-ended questions.
3. With 16,188 images and 136,195 expert-aligned QA pairs in the 3D-RAD-T training set, and 33,910 QA pairs in the benchmark set, the dataset provides a robust foundation.
4. The paper is well-written, organized, and easy to understand.

---

> ### Author Rebuttal · Authors · 2025-07-31
>
> ### 1.The authors are encouraged to revise the dataset name.
>
> Thank you for your valuable suggestion. We will revise the name of our dataset to 3D-CT-RAD in the camera-ready version to better align with our existing data.
> In addition, we consider incorporating other publicly available medical datasets into our framework using the same semi-automated pipeline, either as new tasks or as extensions, to further enrich the diversity of clinical scenarios. We have already incorporated other public medical datasets into our framework as extended sets using the same semi-automated pipeline, enriching the diversity of scenarios[1,2]. Specifically, we introduced two new datasets from additional data sources with more modalities, demonstrating the transferability of our semi-automated annotation and scoring system. We also plan to continue integrating tasks from other datasets.
> Therefore, we believe that 3D-RAD not only fills the gap in existing VQA tasks regarding 3D and temporal modeling, but also provides a reusable and generalizable paradigm for constructing medical VQA benchmarks.
>
> |NewBench|Modality|Cases|QA|Data Source|
> |-|-|-|-|-|
> |BIMCV-1k Bench|CT|218|1308|BIMCV|
> |Rad3D-1k Bench|MRI/X-ray/DSA(angiography)/Ultrasound/Fluoroscopy/Annotated image/Mammography/Nuclear medicine/CT|190|1530|Rad3D|
>
> ### 2. No ablation studies are provided to examine cross-task effectiveness.
> Thank you for pointing out the need for ablation studies. Indeed, we only considered the impact of data scale (1%, 10%, 100%) on model performance (see Table 5 in appendix). We have now added task-level ablation studies to examine cross-task effectiveness. The table below presents results where each task is fine-tuned individually and then evaluated on all other tasks.
> As shown, fine-tuning on a single task can still improve performance on other tasks. However, fine-tuning on the full dataset consistently yields the best results. This demonstrates that the effectiveness of our dataset stems not only from its size but also from the inclusion of high-quality domain knowledge.
>
> Table 5 in appendix
> |Task|Metric|Zero-shot|1%|10%|100%|
> |-|-|-|-|-|-|
> |Task1|BLEU|16.31|20.54|29.35|39.66|
> ||Rouge|23.19|24.63|38.25|50.52|
> ||BERTScore|86.92| 88.23 |89.81|92.19|
> |Task2|BLEU|15.06|21.10|30.54|33.28|
> ||Rouge|23.19|26.03|37.53|42.45|
> ||BERTScore|87.11|88.13|89.78|90.72|
> |Task3|BLEU|2.55|5.65|15.23|20.85|
> ||Rouge|5.63|6.91|23.76|30.38|
> ||BERTScore|85.74|84.89|90.70|93.25|
> |Task4|Acc|40.25|80.85|80.93|82.43|
> |Task5|Acc|24.31|61.01|74.19|74.77|
> |Task6|Acc|25.40|41.17|48.11|49.30|
>
> |Finetuned Task|Evaluate Task4(ACC)|Evaluate Task5(ACC)|Evaluate Task6(ACC)|
> |-|-|-|-|
> |Only Task1|81.88|48.73|73.55|
> |Only Task2|82.03|38.78|72.61|
> |Only Task3|81.94|48.30|72.78|
> |Only Task4|81.96|38.97|73.52|
> |Zero-Shot M3D(Llama2-7B)|18.00|25.47|24.17|
> |Zero-Shot M3D(Phi3-4B)|40.25|25.40|24.31|
> |All Tasks(Ours)|82.43|49.30|74.77|
>
> |Finetuned Task|Evaluate Task2(BLEU/ROUGE1/F1)|
> |-|-|
> |Only Task5|31.60/42.01/90.48|
> |Only Task6|32.45/42.87/90.73|
> |Zero-Shot M3D(Llama2-7B)|9.10/18.64/86.07|
> |Zero-Shot M3D(Phi3-4B)|15.06/23.19/87.11|
> |All Tasks(Ours)|33.28/42.45/90.72|
>
> ### 3. The paper lacks experiments evaluating the transferability of the fine-tuned models to the M3D-VQA dataset.
> Thank you for your valuable suggestion regarding model transferability. Due to licensing restrictions from Radiopaedia, the image component of the M3D dataset has been taken down and access is now prohibited because the scraping process violated usage terms. Out of ethical considerations, we therefore did not include results on this dataset in our paper.
> We appreciate the reviewer’s insightful suggestion regarding model transferability. While our primary contribution focuses on dataset construction (DB track) rather than model development, we fully agree that understanding model generalization is important. Accordingly, we conducted supplementary experiments to explore transferability across tasks, complementing our core emphasis on dataset usability and value: we constructed 1k VQA subsets from two external datasets—BIMCV and Rad3D—covering various modalities (obtained with permission and not subject to ethical concerns). We then evaluated both our model and the original M3D model in a zero-shot setting.
> Results show that even on out-of-domain datasets, our model achieves state-of-the-art performance in the zero-shot setting, demonstrating strong generalization after fine-tuning on our high-quality dataset.
>
> |BIMCV-1k Bench|Image Observation(BLEU/ROUGE1/F1) |Anomaly Detection(BLEU/ROUGE1/F1)
> |-|-|-|
> |RadFM|14.37/14.58/86.96|13.71/10.45/86.29|
> |M3D(Llama2-7B)|9.89/13.38/85.67|9.20/8.75/84.78|
> |M3D(Phi3-4B)|10.04/12.84/85.66|8.72/9.82/84.89|
> |Ours-Finetuned(Llama2-7B)|22.97/16.06/87.06|22.26/10.59/86.63|
> |Ours-Finetuned(Phi3-4B)|14.05/19.42/86.98|9.31/13.08/85.97|
>
> |Rad3D-1k Bench|Image Observation(BLEU/ROUGE1/F1) |Anomaly Detection(BLEU/ROUGE1/F1)
> |-|-|-|
> |RadFM|15.22/13.89/86.43|12.86/10.83/85.61|
> |M3D(Llama2-7B)|12.62/14.38/85.05|8.64/9.32/84.11|
> |M3D(Phi3-4B)|11.18/12.29/84.89|8.78/9.87/84.29|
> |Ours-Finetuned(Llama2-7B)|21.28/10.02/86.26|20.29/8.35/85.66|
> |Ours-Finetuned(Phi3-4B)|12.71/14.82/85.89|6.90/8.71/84.78|
>
> References:
> *[1]Chen Y, Liu C, Liu X, et al. Bimcv-r: A landmark dataset for 3d ct text-image retrieval*
> *[2]Wu C, Zhang X, Zhang Y, et al. Towards generalist foundation model for radiology by leveraging web-scale 2d&3d medical data*

---

> > ### Comment · Reviewer_e2md · 2025-08-04
> > **Good Rebuttal**
> >
> > The reviewer thanks the authors for addressing all of the raised concerns.

---

> > > ### Author Response · Authors · 2025-08-04
> > >
> > > We are pleased that our rebuttal has addressed your concerns. We sincerely appreciate your recognition of our work and the efforts during the rebuttal process. Your feedback has been invaluable in helping us further improve the manuscript. Thank you again for your time and thoughtful review.

---

### Official Review · Reviewer_h2tY · 2025-07-01

**Rating:** 4
**Confidence:** 4

**Summary:**

The paper introduces 3D-RAD, 3D medical visual question answering (Med-VQA). It expands six tasks from CT-RATE dataset and uses existing VLMs to benchmark. Results show that VLMs perform poorly on multi-temporal tasks, and fine-tuning on the dataset shows performance improvement.

**Dataset Code Accessibility:**

Yes

**Ethical Considerations:**

No, there are no or only very minor ethics concerns

**Final Justification:**

Mostly I had issues with the rehashing CT-RATE dataset but given the rebuttal and revision, there are some merits for the clinically relevant tasks and evaluation. Thus I give score 4

**Limitations Weaknesses:**

- It's mostly rehashing CT-RATE where image-report pairs come from an existing public dataset. No new raw clinical data was collected
- Temporal modeling is superficial: Task 6 provides only binary historical labels. This is a major simplification
- Medical Computation (Task 3) relies on extracting numerical values from reports not direct measurement from images. This is a major concern for visual reasoning
- Tasks like “Static Temporal Diagnosis” ask the model to infer longitudinal progression from a single time point which is pretty artificial
- Tasks, prompts, and evaluations are highly structured and templated and there's no free-text to represent real medical queries

**Strengths Contributions:**

- It recognize the clearly underserved area for 3D medical imaging for VQA style benchmarks
- The evaluation is systematic, the baseline models are diverse, the experiments cover enough technical details
- Some novel tasks requiring longitudinal reasoning

---

> ### Author Rebuttal · Authors · 2025-07-31
>
> ### 1. No new raw clinical data was collected.
> Thank you for your concerns. Our work focuses on systematic reconstruction, targeted selection, and deep information mining for multi-task medical VQA. We addressed key limitations in existing datasets, especially under limited information conditions:
> * Most medical VQA datasets (e.g., VQA-RAD, PathVQA) rely on 2D static images without 3D or temporal modeling (Table 1). While CT-RATE offers 3D multi-phase scans, its temporal structure is not explicitly modeled or utilized.
> * Small scale and weak quality control: Existing datasets lack scale and systematic diversity. We created six tasks with over 160k QA pairs and a rigorous quality control pipeline (Section 3.3), significantly enhancing data quality.
> * Field advancement and generalizability: Due to ethical and privacy constraints, efficiently reusing existing datasets is both practical and necessary. Our semi-automated pipeline, effective on CT-RATE and generalizable to other clinical datasets, supports scalable, standardized multi-modal data construction (Fig. 3). All construction code will be released.
>
> We recognize the value of diverse data for improving model generalizability. 3D-RAD builds on CT-RATE[1], the only public dataset with high-quality 3D images, paired reports, and well-annotated multi-phase studies. Covering ages 18–102 and a range of cardiopulmonary conditions, CT-RATE offers varied slice numbers, thicknesses, pixel spacing, and CT vendors—providing a strong foundation for 3D medical VQA.
> Expanding on CT-RATE, 3D-RAD includes over ten organs and nearly twenty disease types. It is large-scale, diverse, and rich in structured, filterable data. To overcome the limitations of a single source, we propose a six-task VQA framework that fully leverages this diversity.
> We have extended our semi-automated pipeline\[2,3] to include other public medical datasets, demonstrating the transferability of our annotation and scoring system across modalities. More datasets will be integrated. As such, 3D-RAD not only fills the gap in 3D and temporal VQA but also offers a reusable, generalizable benchmark construction paradigm.
>
> |NewBench|Modality|Cases|QA|Data Source|
> |-|-|-|-|-|
> |BIMCV-1k Bench|CT|218|1308|BIMCV|
> |Rad3D-1k Bench|MRI/X-ray/DSA(angiography)/Ultrasound/Fluoroscopy/Annotated image/Mammography/Nuclear medicine/CT|190|1530|Rad3D|
>
> |BIMCV-1k Bench|Image Observation(BLEU/ROUGE1/F1) |Anomaly Detection(BLEU/ROUGE1/F1)
> |-|-|-|
> |RadFM|14.37/14.58/86.96|13.71/10.45/86.29|
> |M3D(Llama2-7B)|9.89/13.38/85.67|9.20/8.75/84.78|
> |M3D(Phi3-4B)|10.04/12.84/85.66|8.72/9.82/84.89|
> |Ours-Finetuned(Llama2-7B)|22.97/16.06/87.06|22.26/10.59/86.63|
> |Ours-Finetuned(Phi3-4B)|14.05/19.42/86.98|9.31/13.08/85.97|
>
> |Rad3D-1k Bench|Image Observation(BLEU/ROUGE1/F1) |Anomaly Detection(BLEU/ROUGE1/F1)
> |-|-|-|
> |RadFM|15.22/13.89/86.43|12.86/10.83/85.61|
> |M3D(Llama2-7B)|12.62/14.38/85.05|8.64/9.32/84.11|
> |M3D(Phi3-4B)|11.18/12.29/84.89|8.78/9.87/84.29|
> |Ours-Finetuned(Llama2-7B)|21.28/10.02/86.26|20.29/8.35/85.66|
> |Ours-Finetuned(Phi3-4B)|12.71/14.82/85.89|6.90/8.71/84.78|
>
> ### 2. Temporal modeling is superficial
> Thank you for your valuable suggestion. Our choice is driven by several practical considerations:
> * Model compatibility: Current medical VLMs (e.g., RadFM, OmniV, M3D) support only single-phase 3D inputs. To maintain fair evaluation, we introduce historical context via textual labels, enabling temporal reasoning within existing architectural limits.
> * Evaluability considerations: We previously explored multi-phase reports, but this led to unstable, non-unique responses due to the open-ended nature of generative models and mismatched question lengths. In contrast, binary classification provides a closed-question setup, offering clearer and more consistent evaluation of temporal understanding. Several medical VQA datasets, including VQA-RAD and Slake, also use binary answers (Yes/No) to represent pathologies or anatomical structures, enabling straightforward supervision and improving model performance.
> * Future readiness: As noted in our Limitations, the dataset is designed for future expansion. Once VLMs support multi-phase 3D inputs, binary labels can be directly replaced with image-based comparisons, requiring no structural overhaul.
>
> Thus, Task 6 balances current model constraints with forward-compatible design and reliable evaluation.
>
> ### 3. Task 3 relies on extracting numerical values from reports.
> Thank you for your insightful comment. The extracted numerical values are only used for supervision during training; at inference time, the model receives 3D images and natural language questions, and must infer numerical answers from visual input alone.
> Task 3 focuses on quantitative reasoning over 3D medical images, using radiologist-reported values as clinically grounded supervision. It tests the model’s ability to align visual features with numerical understanding, serving as a key component of medical VQA \[4].
> Our results show that even fine-tuned models struggle with such computations, underscoring the task’s difficulty and the value of our dataset. The richer 3D data, however, offers room for future models to improve.
>
> Table 4
> |Model|BLEU|Rouge|BERTScore|
> |-|-|-|-|
> |RadFM|3.34|6.62|86.85|
> |M3D(Llama2-7B)|15.95|23.24|91.50|
> |M3D(Phi3-4B)|2.55|5.63|85.74|
> |OmniV(Qwen2.5-1.5B)|2.52|7.88|85.66|
>
> ### 4. “Static Temporal Diagnosis” requires inferring progression from a single time point
> Thank you for your insightful concerns. In clinical practice, physicians often infer prior conditions from single timepoint images—such as fibrosis post-pneumonia, calcifications from past tuberculosis, or structural changes after surgery or radiotherapy[5,6]—to understand disease progression.
> Studies also show that residual signs in current images(medical history) can indicate future risks, prompting increased vigilance for recurrence[7,8]. Task 4 emulates this reasoning process, addressing models' limitations in temporal understanding and disease trajectory assessment.
> We believe this task holds strong clinical relevance, offering a novel way to evaluate longitudinal reasoning and promoting physician-like interpretability and applicability.
> Our experiments confirm that this task is challenging for current models, as they lack the ability to infer from subtle image cues. No existing benchmarks or datasets effectively assess or support improvement in this area. However, as noted above, the task is clinically important—one of the key motivations behind our dataset—which offers a valuable opportunity for advancing models' capabilities in this complex aspect.
>
> ### 5. Tasks, prompts, and evaluations are overly templated, lacking free-text to reflect real medical queries.
> Thank you for raising this important point. We followed the design principles of other datasets and have incorporated a wider variety of question types and coverage than existing related datasets[9].
> Moreover, during construction, we used a strategy of "precise clinical text guidance + diverse free-text reconstruction by large models," ensuring accurate QA content and diverse questions.
>
> #### Open question:
> * **Free-form Rewriting with LLMs**: To ensure semantic accuracy while enhancing naturalness, we employ GPT to freely rewrite structured clinical questions into semantically equivalent free-form queries. Through prompt engineering, we align the questions more closely with how clinicians naturally ask them in real-world settings.
> * **Diverse question starters**: To reduce repetitive patterns, we adopted the “6W” framework and, following expert validation, selected three clinically common question `<Starter>`—*what*, *where*, and *which*—for our templates.
> * **Randomized starter assignment**: Each `<Starter>` appears twice to form a list of six, which is shuffled and assigned per report to ensure diversity in `<Starter>`, image content, and textual cues. This strategy promotes uniqueness and balanced distribution across the dataset (see Fig. 9).
> * **Entity coverage analysis**: We applied named entity recognition to assess anatomical entity distribution in the questions. Statistics are reported in Fig. 5 and Section 3.3, with Fig. 27 (appendix) providing task-wise word clouds that further demonstrate the dataset’s organ and expression diversity.
>
> #### Closed question:
> * **Carefully designed prompt templates**: 20 diverse templates per closed question task were crafted by experts to enhance linguistic variety and reduce repetition (Fig. 12–13).
> * **Multi-clause Question Generation for Multi-phase Tasks**:  For Tasks 5 and 6, questions are composed of sub-clauses (sequence, fluctuation, and query patterns). We designed 20 templates per sub-clause and randomly combined them to enrich expression diversity (Fig. 14).
>
> References:
> *[1]Hamamci I E, Er S, Wang C, et al. Developing generalist foundation models from a multimodal dataset for 3d computed tomograph*
> *[2]Chen Y, Liu C, Liu X, et al. Bimcv-r: A landmark dataset for 3d ct text-image retrieval*
> *[3]Wu C, Zhang X, Zhang Y, et al. Towards generalist foundation model for radiology by leveraging web-scale 2d&3d medical data*
> *[4]Khandekar N, Jin Q, Xiong G, et al. Medcalc-bench: Evaluating large language models for medical calculations*
> *[5]Brixey A G, Oh A S, Alsamarraie A, et al. Pictorial review of fibrotic interstitial lung disease on high-resolution CT scan and updated classification*
> *[6]Chae K J, Hwang H J, Duarte Achcar R, et al. Central role of CT in management of pulmonary fibrosis*
> *[7]Meghji J, Simpson H, Squire S B, et al. A systematic review of the prevalence and pattern of imaging defined post-TB lung disease*
> *[8]Seon H J, Kim Y I, Lim S C, et al. Clinical significance of residual lesions in chest computed tomography after anti-tuberculosis treatment*
> *[9]Bai F, Du Y, Huang T, et al. M3d: Advancing 3d medical image analysis with multi-modal large language models*

---

> > ### Comment · Reviewer_h2tY · 2025-08-06
> >
> > I have read authors' rebuttal. There's still concern about new raw clinical data but given the new task/eval angles, I'm raising rating to 4

---

> > > ### Author Response · Authors · 2025-08-06
> > >
> > > Thank you very much for recognizing our work and reconsidering your rating. This paper presents the first comprehensive dataset for multi-task and multi-temporal reasoning in 3D medical imaging, aiming to advance 3D medical VLMs. Given the scalability of our approach, we are committed to maintaining and expanding 3D-RAD as a lasting contribution to the community. We sincerely appreciate your time and valuable feedback.

---

> ### Author Response · Authors · 2025-08-04
>
> Dear Reviewer h2tY,
>
> We sincerely appreciate your thoughtful and thorough review of our work. We have carefully addressed the key concerns you raised and made corresponding revisions. As the discussion period advances and the remaining time gradually narrows, we hope our responses align with your expectations. If there are any further questions or clarifications that may assist you in finalizing your evaluation, we would be more than happy to provide them.

---

> ### Author Response · Authors · 2025-08-06
> **Follow-up Response**
>
> Dear Reviewer h2tY,
>
> Thank you again for your thoughtful review. We have addressed all concerns from other reviewers, and have carefully responded to each of your comments. We would be grateful for your feedback.
>
> We truly value the opportunity to engage in this discussion and would be happy to clarify any remaining issues. We sincerely appreciate your time as the discussion period progresses.
>
> Best regards,
>
> the authors

---

### Official Review · Reviewer_R2sE · 2025-07-02

**Ethics Flags:** Safety and security, Environmental im…
**Rating:** 6
**Confidence:** 4

**Summary:**

This paper presents a new benchmark 3D-RAD for 3D Med-VQA with CT scans. It contains six diverse VQA tasks: anomaly detection, image observation, medical computation, existence detection, static temporal diagnosis, and longitudinal temporal diagnosis, supporting both open- and closed-ended questions. The new benchmark poses challenges on computational tasks and multi-stage temporal analysis. Preliminary results on med-VLMs show that existing models are still limited in 3D understanding. This paper also present a new training dataset which could help advance performance of med-VLM for 3D VQA and understanding. The dataset and code are publicaly available and comply with the ethics considerations.

**Dataset Code Accessibility:**

Yes

**Dataset Code Comments:**

Overall they are good. Code could be further clarified with more comments.

**Ethical Comments:**

The paper examined the licenses of collected datasets, and comply with them accordingly.

**Ethical Considerations:**

No, there are no or only very minor ethics concerns

**Final Justification:**

This paper makes a meaningful and timely contribution by introducing a 3D Med-VQA benchmark and training set, which could help advance research in medical vision-language understanding. During the rebuttal, they add more 3D benchmarks, as well as including multiple new benchmarks. My major concerns are mostly addressed. Therefore, I have raised my initial score.

**Limitations Weaknesses:**

I have the following concerns and quesitons:
- Is it possible to include other 3D CT scans rather than CT-RATE, either for building the benchmark or training set? Or is there an effective automatic pipeline to generalize to more 3D datasets in the future to scale up this benchmark?
- Is it possible to include more baselines, thought they might not be tailored for 3D understanding. Including the performance of such generic or medical VLMs could help better understand the limitations of existing works.
- In table 2, it should be Task 1-6 rather than task 2-7.

**Strengths Contributions:**

- This work is very timing to the community. While there are lots of 2D VQA benchmarks in medical image understanding, a systematic 3D benchmark is very appealing to researchers, which could be leveraged to support more diverse clinical scenarios.
- This work provide a new benchmark as well as a high-quality training set to enhance and evluate the performance of medical VLMs. The benchmark support 6 distinct tasks for 3D understanding and temporal analysis, including both open and close-end questions.
- The authors conducted a semi-automated scoring and filtering pipeline. Both GPTs and human evaluations are employed to enforce data quality.
The preliminary results showcase the limitations of existing 3D med-VLMs and the proposed training set can significantly improve model performance.

---

> ### Author Rebuttal · Authors · 2025-07-31
>
> ### 1. Is it possible to include other 3D CT scans rather than CT-RATE, either for building the benchmark or training set? Or is there an effective automatic pipeline to generalize to more 3D datasets in the future to scale up this benchmark?
> Thank you for your suggestion regarding the inclusion of additional 3D CT datasets. Our current “semi-automated” data construction pipeline is centered on: using LLMs to automatically generate candidate QA pairs for each task; manual review and filtering; and supporting rapid adaptation to new data sources. This provides a solid foundation for incorporating more 3D CT data in the future.
> In addition, we consider incorporating other publicly available medical datasets into our framework using the same semi-automated pipeline, either as new tasks or as extensions, to further enrich the diversity of clinical scenarios. We have already incorporated other public medical datasets into our framework as extended sets using the same semi-automated pipeline, enriching the diversity of scenarios[1,2]. Specifically, we introduced two new datasets from additional data sources with more modalities, demonstrating the transferability of our semi-automated annotation and scoring system. We also plan to continue integrating tasks from other datasets.
> Therefore, we believe that 3D-RAD not only fills the gap in existing VQA tasks regarding 3D and temporal modeling, but also provides a reusable and generalizable paradigm for constructing medical VQA benchmarks.
>
> |NewBench|Modality|Cases|QA|Data Source|
> |-|-|-|-|-|
> |BIMCV-1k Bench|CT|218|1308|BIMCV|
> |Rad3D-1k Bench|MRI/X-ray/DSA(angiography)/Ultrasound/Fluoroscopy/Annotated image/Mammography/Nuclear medicine/CT|190|1530|Rad3D|
>
> |BIMCV-1k Bench|Image Observation(BLEU/ROUGE1/F1) |Anomaly Detection(BLEU/ROUGE1/F1)
> |-|-|-|
> |RadFM|14.37/14.58/86.96|13.71/10.45/86.29|
> |M3D(Llama2-7B)|9.89/13.38/85.67|9.20/8.75/84.78|
> |M3D(Phi3-4B)|10.04/12.84/85.66|8.72/9.82/84.89|
> |Ours-Finetuned(Llama2-7B)|22.97/16.06/87.06|22.26/10.59/86.63|
> |Ours-Finetuned(Phi3-4B)|14.05/19.42/86.98|9.31/13.08/85.97|
>
> |Rad3D-1k Bench|Image Observation(BLEU/ROUGE1/F1) |Anomaly Detection(BLEU/ROUGE1/F1)
> |-|-|-|
> |RadFM|15.22/13.89/86.43|12.86/10.83/85.61|
> |M3D(Llama2-7B)|12.62/14.38/85.05|8.64/9.32/84.11|
> |M3D(Phi3-4B)|11.18/12.29/84.89|8.78/9.87/84.29|
> |Ours-Finetuned(Llama2-7B)|21.28/10.02/86.26|20.29/8.35/85.66|
> |Ours-Finetuned(Phi3-4B)|12.71/14.82/85.89|6.90/8.71/84.78|
>
> ### 2. Is it possible to include more baselines, thought they might not be tailored for 3D understanding. Including the performance of such generic or medical VLMs could help better understand the limitations of existing works.
> Thank you for your thoughtful suggestion. We constructed a new 2D-Bench by extracting 2D slices from 3D images and evaluated several general-purpose VLMs on it. As shown, even the latest GPT-4.1 performs significantly worse with 2D inputs compared to models using 3D inputs (despite the latter having far fewer parameters), highlighting the importance of leveraging 3D data.
>
> |Model|Task1(BLEU/ROUGE1/F1)|Task2(BLEU/ROUGE1/F1)|Task3(BLEU/ROUGE1/F1)|Task4(ACC)|Task5(ACC)|Task6(ACC)|
> |-|-|-|-|-|-|-|
> |qwen2.5-vl(7B)|2.70/ 28.83/88.68|2.12/25.51/88.15|1.84/27.18/93.14|71.31|38.30|38.24|
> |intern-vl-3(8B)|1.41/33.35/89.41|1.41/29.77/89.02|2.10/22.34/92.76|78.18|14.74|14.54|
> |gemma-3(12b)|1.50/18.80/87.16|1.76/19.43/87.23|0.37/3.95/86.13|71.37|24.09|24.09|
> |gemini-2.0-flash|10.05/7.83/85.49|7.49/6.88/85.09|0.25/0.06/82.81|52.77|18.97|17.22|
> |gpt-4.1-nano|7.34/11.25/85.40|6.24/11.05/85.30|1.93/5.63/83.96|49.94|23.61|31.28|
> |gpt-4.1-mini|7.74/11.91/85.62|7.22/11.41/85.64|3.09/6.20/84.26|78.12|35.54|66.66|
> |gpt-4.1|6.83/9.96/85.42|6.14/8.86/85.24|0.24/0.24/81.39|78.24|42.65|68.05|
> |Ours-Finetuned(Llama2-7B)|31.28/39.12/90.00|25.25/33.76/89.16|30.54/36.06/94.65|81.09|51.20|74.78|
> |Ours-Finetuned(Phi3-4B)|39.66/50.52/92.19|33.28/42.45/90.72|20.85/30.38/93.25|82.43|49.30|74.77|
>
> ### 3. In table 2, it should be Task 1-6 rather than task 2-7.
> Thank you for your careful review and correction. Apologies for the mistake — it should indeed be “Task 1–6” instead of “Task 2–7.” We will fix this in the camera-ready version.
>
> References:
> *[1]Chen Y, Liu C, Liu X, et al. Bimcv-r: A landmark dataset for 3d ct text-image retrieval*
> *[2]Wu C, Zhang X, Zhang Y, et al. Towards generalist foundation model for radiology by leveraging web-scale 2d&3d medical data*

---

> > ### Comment · Reviewer_R2sE · 2025-08-01
> >
> > Thanks for the responses. I appreciate the authors' efforts to add more 3D benchmarks, as well as including multiple new benchmarks. My major concerns are mostly addressed. I will consider increasing my score.

---

> > > ### Author Response · Authors · 2025-08-01
> > >
> > > Thank you for your positive feedback and for recognizing our efforts and contributions to 3D medical benchmarks. We’ll include the additional content in the camera-ready version. We truly appreciate your acknowledgment and your consideration of a higher score. Thank you again for your constructive suggestions.

---

### Official Review · Reviewer_q8Fu · 2025-07-03

**Rating:** 4
**Confidence:** 4

**Summary:**

This paper presents 3D-RAD, a large-scale dataset designed for effective training and benchmarking of 3D Med-VQA tasks using radiology CT and text. The 3D-RAD consists of 34k data entries, covering 18 diseases across six challenging VQA task types, spanning open-ended and closed-ended tasks. This paper further releases a high-quality training dataset of 136K samples, which can be used to fine-tune state-of-the-art 3D vision-language models and effectively improve their performance.

**Dataset Code Accessibility:**

Yes

**Dataset Code Comments:**

The GitHub link provides the code with clear instructions on how to use the code.

**Ethical Comments:**

There is no significant ethical concern because the proposed dataset is built upon publicly available datasets whose licenses(CC BY-NC-SA) allow the redistribution of their data.

**Ethical Considerations:**

No, there are no or only very minor ethics concerns

**Final Justification:**

The rebuttal has addressed most of my concerns, e.g., the processing details, quality control, etc. As such, I incline to upgrade my rating.

**Limitations Weaknesses:**

1. More details about 18 binary classification prompts and the predefined abnormality labels (Line 202) should be given.

2. Line 236, please detail how to perform automatic deduplication, scoring, and filtering.

3. The automatic scoring and manual sampling-based validation (Line 198-199) should be further clarified.

4. As a benchmark, only evaluating three VLMs (RadFM, M3D, and OminV) on 3D-RAD is not enough. Please consider including more VLMs.

5. 3D-RAD is built upon a single dataset, CT-RATE, which may limit its diversity of data sources.

6. The training set has 136K samples, but lines 55 and 125 state that the final dataset has only 34K QA pairs. How to the 136K high-quality training set from 34K pairs? This can be confusing because the 3D-RAD-Bench and 3D-RAD-T have not been introduced in the Introduction.

7. Why are five words chosen to constrain generated answers rather than six, seven or ten words? (Minor concern)

**Strengths Contributions:**

+ This paper introduces 3D-RAD, a large-scale dataset designed for effective training and benchmarking of 3D Med-VQA tasks using radiology CT and text. It has 34k data entries, covers over 18 diseases, and supports advanced open-ended and closed-ended reasoning tasks, such as medical computation and temporal analysis. This dataset can contribute to the research on 3D medical vision-language models.
+ This paper further presents a training subset, 3D-RAD-T, for model fine-tuning. The fine-tuned models show clear performance gains.
+ This paper reveals critical limitations, e.g., complex multi-temporal reasoning tasks, in current 3D vision-language models, showing a future direction for improving these models.

---

> ### Author Rebuttal · Authors · 2025-07-31
>
> ### 1.Clarify the 18 binary prompts and predefined abnormality labels.
> Thank you for your valuable suggestion. In Fig. 12–14 of the Appendix, we provide full prompts and detailed labels. The 18 question aspects cover a wide range of thoracic conditions, including vascular and cardiac abnormalities, lung parenchymal changes, pleural findings, and other structural or pathological features.
> We will include this important content in the main text of the camera-ready version.
> ### 2.Describe methods for automatic deduplication, scoring, and filtering.
> Thank you for pointing out the need to elaborate on our data processing pipeline. We used semi-automated strategies to ensure question diversity, answer quality, and reduce redundancy:
> #### Open question deduplication:
> * **Diverse question starters**: To reduce repetitive patterns, we adopted the "6W"[1,2] framework and, following expert validation, selected three clinically common question `<Starter>`*"what, where, and which"* for our templates.
> * **Randomized starter assignment**: Each `<Starter>` appears twice to form a list of six, which is shuffled and assigned per report to ensure diversity in `<Starter>`, image content, and textual cues. This strategy promotes uniqueness and balanced distribution across the dataset (see Fig. 9).
> * **High-frequency sample truncation**: For questions or answers appearing over 10 times, only the first 10 are retained. This reduces redundancy while preserving essential semantic and contextual diversity, better reflecting common clinical inquiries than extreme deduplication.
> * **Entity coverage analysis**: We applied named entity recognition to assess anatomical entity distribution in the questions. Statistics are reported in Fig. 5 and Section 3.3, with Fig. 27 (appendix) providing task-wise word clouds that further demonstrate the dataset’s organ and expression diversity.
>
> #### Closed question deduplication:
> * **Carefully designed prompt templates**: 20 expert-crafted templates were created per closed question task to ensure linguistic variety and minimize repetition (Fig. 12–13).
> * **Multi-clause Question Generation for Multi-phase Tasks**: For Tasks 5 and 6, questions consist of sub-clauses—sequence, fluctuation, and query patterns—with 20 templates each (Fig. 14), randomly combined to boost diversity and expression range.
> #### Scoring, and Filtering：
> To ensure scoring consistency, we proportionally sample 600 QA pairs across six tasks. Four models independently rate each pair using a five-dimension rubric. We compute average scores, rank all pairs, and assess inter-model agreement using four metrics. GPT-4o-mini shows the highest consistency and is chosen for final scoring.
> A semi-automated pipeline maintains 3D-RAD quality: each QA is rated 1–5 across five dimensions. Pairs scoring below 3 in any dimension or on average are removed. The rest are ranked by average score, with top entries forming the final dataset. The full prompt appears in Fig. 15.
>
> In Supplementary Material A.12, in addition to the above steps, we provide more detailed information.
> ### 3.Clarified automatic scoring and manual sampling-based validation.
> Thank you for your request regarding the scoring and validation process. Annotators followed a detailed guide, using two reference examples and sampled QA pairs with clinical reports. Each QA was rated on five dimensions (0–5) and labeled for binary consistency with the source report (1 = consistent, 0 = inconsistent).
> To ensure reliability, we identified 53 inconsistent QAs (consistency = 0), then retained only those with an average score ≥ 3 and all dimensions ≥ 3, resulting in 23 low-quality samples. This yields a factual consistency rate of 96.17%.
> Given the dataset size (170K QAs), our sampling and validation strategy ensures both efficiency and benchmark-level quality.
>
> In Supplementary Material A.13, in addition to the above steps, we provide more detailed information.
> ### 4.Consider including more VLMs.
> Thank you for the valuable suggestion on exploring more VLMs. While VLMs capable of processing 3D medical inputs remain rare, our evaluation includes representative state-of-the-art models in this space. This underscores the importance of 3D-RAD, which offers standardized, diverse tasks to advance 3D medical understanding in both general and domain-specific VLMs.
> We have added zero-shot evaluations using general VLMs. Though some models perform better—especially on tasks 4–6 with more reliable metrics—none achieve strong results across all tasks. Models fine-tuned on 3D-RAD consistently outperform general models, highlighting the dataset’s value in promoting domain-specific learning.
>
> |Model|Task1(ROUGE1/F1)|Task2(ROUGE1/F1)|Task3(ROUGE1/F1)|Task4(ACC)|Task5(ACC)|Task6(ACC)|
> |-|-|-|-|-|-|-|
> |llava-onevision(7b)|26.30/86.79|22.34/86.24|5.93/91.81|29.67|6.86|6.86|
> |qwen2.5-vl(3B)|22.37/83.77|24.40/84.96|18.21/92.31|22.75|24.09|24.09|
> |qwen2.5-vl(32B)|20.30/87.64|21.77/87.66|11.46/90.73|58.60|28.62|28.36|
> |gemma-3(4b)|22.50/87.41|22.91/87.44|8.56/90.50|21.47|10.75|10.74|
> |gemma-3(27b)|30.43/88.89|26.21/88.15|10.55/90.54|28.57|24.09|24.09|
> |gemini-2.0-flash|27.93/89.29|24.70/88.93|0.53/84.36|40.42|23.08|21.33|
> |intern-vl-3(8B)|34.31/89.63|28.44/88.83|10.65/91.64|55.30|26.63|26.61|
> |gpt-4.1-nano|11.41/85.32|10.99/85.00|4.31/83.63|37.95|19.95|25.11|
> |gpt-4.1|16.55/86.50|15.08/86.03|2.76/82.82|61.98 |34.08|68.05|
> |Ours-Finetuned(Llama2-7B)|39.12/90.00|33.76/89.16|36.06/94.65|81.09|51.20|74.78|
> |Ours-Finetuned (Phi3-4B)|50.52/92.19|42.45/90.72|30.38/93.25|82.43|49.30|74.77|
> ### 5. Limited diversity of data sources.
> Thank you for emphasizing the importance of data diversity. We fully agree that it plays a key role in enhancing model generalizability. 3D-RAD builds upon CT-RATE, the only public dataset offering high-quality 3D images, paired reports, and annotated multi-phase studies—covering a wide age range, various pathologies, multiple CT vendors, and a balanced sex distribution \[3].
> CT-RATE covers over ten organs and nearly twenty disease types. It is large-scale, diverse, and rich in filterable information. To mitigate the limitations of a single source, we designed a multi-task, multi-dimensional VQA framework that fully exploits the inherent diversity of CT-RATE.
>
> Due to ethical and privacy constraints in medical data collection, maximizing public datasets is both practical and necessary. Our pipeline, validated on CT-RATE, generalizes well to other clinical datasets, enabling standardized multimodal data construction (Fig. 3). We have integrated additional public datasets using the same semi-automated approach to enhance modality and scenario diversity [4,5], demonstrating the transferability of our annotation and scoring system, with more integrations planned.
>
> |NewBench|Modality|Cases|QA|Data Source|
> |-|-|-|-|-|
> |BIMCV-1k Bench|CT|218|1308|BIMCV|
> |Rad3D-1k Bench|MRI/X-ray/DSA(angiography)/Ultrasound/Fluoroscopy/Annotated image/Mammography/Nuclear medicine/CT|190|1530|Rad3D|
>
> |BIMCV-1k Bench|Image Observation(BLEU/ROUGE1/F1) |Anomaly Detection(BLEU/ROUGE1/F1)
> |-|-|-|
> |RadFM|14.37/14.58/86.96|13.71/10.45/86.29|
> |M3D(Llama2-7B)|9.89/13.38/85.67|9.20/8.75/84.78|
> |M3D(Phi3-4B)|10.04/12.84/85.66|8.72/9.82/84.89|
> |Ours-Finetuned(Llama2-7B)|22.97/16.06/87.06|22.26/10.59/86.63|
> |Ours-Finetuned(Phi3-4B)|14.05/19.42/86.98|9.31/13.08/85.97|
>
> |Rad3D-1k Bench | Image Observation (BLEU / ROUGE1 / F1) | Anomaly Detection (BLEU / ROUGE1 / F1)
> |-|-|-|
> |RadFM|15.22/13.89/86.43|12.86/10.83/85.61|
> |M3D(Llama2-7B)|12.62/14.38/85.05|8.64/9.32/84.11|
> |M3D(Phi3-4B)|11.18/12.29/84.89|8.78/9.87/84.29|
> |Ours-Finetuned(Llama2-7B)|21.28/10.02/86.26|20.29/8.35/85.66|
> |Ours-Finetuned(Phi3-4B)|12.71/14.82/85.89|6.90/8.71/84.78|
> ### 6. Unclear definitions of dataset size.
> Thank you for pointing out this confusion. The discrepancy stems from insufficient clarification in the Introduction, which we will revise. The 3D-RAD dataset comprises two distinct subsets:
> * 3D-RAD-Bench: a benchmark set with ~34K QA pairs spanning six representative tasks, designed for standardized evaluation;
> * 3D-RAD-T (Train): ~136K QA pairs constructed independently from the benchmark using the same procedure, without overlap, expansion, or resampling.
>
> The training set supports model learning with broad coverage, and the benchmark is curated for fine-grained evaluation. We will make this distinction explicit in the camera-ready version.
> ### 7. Five words chosen to constrain generated answers.
> Thank you for raising these critical concerns. We referred to mainstream medical VQA datasets and found that most answers fall within 5 words. Based on experiments with different length limits (1, 3, 5, 10 words) and expert evaluation, we observed that:
> * ~5 words suffice for typical medical VQA answers (e.g., “no obvious lesion”, “new nodule in right lung”);
> * Longer answers (e.g., over 10 words) tend to introduce redundancy and resemble instruction-tuning style responses, deviating from the concise format expected in VQA;
> * A max 5-word constraint offers a balance between clinical precision and brevity.
>
> |Dataset|Typical Length|Example|
> |-|-|-|
> |VQA-RAD|~1-3 words|"hemorrhage""abdomen and pelvis"|
> |SLAKE|~1-4 words|"Liver""Large Bowel"|
> |PathVQA|~1-5 words|"gastrointestinal""in the canals of hering"|
> |M3D-VQA|~1-5 words|"Left kidney""Within the sella turcica"|
>
> References:
> *[1]Antol S et al. Vqa: Visual question answering*
> *[2]Fukui A et al. Multimodal compact bilinear pooling for visual question answering and visual grounding*
> *[3]Hamamci I E et al. Developing generalist foundation models from a multimodal dataset for 3d computed tomograph*
> *[4]Chen Y et al. Bimcv-r: A landmark dataset for 3d ct text-image retrieval*
> *[5]Wu C et al. Towards generalist foundation model for radiology by leveraging web-scale 2d&3d medical data*

---

> > ### Comment · Reviewer_q8Fu · 2025-08-04
> > **Further discussion**
> >
> > Many thanks for the response, which addresses most of my concerns. There are some concerns for further clarification:
> >
> > For Q2, “only the first 10 are retained”: how to ensure the quality of these 10? “We applied named entity recognition to assess anatomical entity distribution in the questions”: what algorithm or strategy is used to recognize the entity?
> > “each QA is rated 1–5 across five dimensions”: I suppose the five dimensions are Visual Verifiability, Specificity & Clarity, Answer Appropriateness, Q-A Alignment, and Linguistic Quality. However, I did not find detailed standards for these five dimensions. For instance, how to rate it according to Q-A Alignment and Linguistic Quality? Any specific instructions to guide how to score across these five dimensions?
> >
> > For Q3, “using two reference examples”: perhaps provide more information regarding the reference examples?

---

> > > ### Author Response · Authors · 2025-08-05
> > > **Response to Further Discussion (Part 1)**
> > >
> > > We sincerely appreciate your detailed and insightful review. It is a pleasure to engage in further discussion with you.
> > >
> > > > how to ensure the quality of these 10?
> > >
> > > The high quality of these top 10 QA pairs is ensured through **a stepwise filtering and prioritization process** based on multi-dimensional scoring. Specifically, we first eliminate any QA pair scoring below 3 in any of the five evaluation dimensions to maintain a baseline quality threshold. The remaining pairs are ranked in descending order of their overall scores.
> > >
> > > If a Question or Answer appears multiple times, we retain only **the top 10 instances with the highest scores**. This scoring-based prioritization guarantees that the selected samples are not only among the best-rated but also varied in context and phrasing.
> > >
> > > After removing duplicates, we **further apply descending score-based selection** to the entire pool of over 400k generated samples, ensuring that the final dataset retains only the most reliable and diverse QA pairs for downstream tasks.
> > >
> > > > what algorithm or strategy is used to recognize the entity?
> > >
> > > Thank you for your attention to this point. As noted in Lines 241–244, we use the **en_ner_bionlp13cg_md** model [1], trained on biomedical literature, to extract anatomical and pathological entities that align with our dataset categories.
> > >
> > > *References:
> > > [1]Neumann M et al. ScispaCy: Fast and Robust Models for Biomedical Natural Language Processing.*
> > >
> > > > Detail standards for these five dimensions.
> > >
> > > Thank you for your careful review. The five dimensions are indeed Visual Verifiability, Specificity & Clarity, Answer Appropriateness, Q-A Alignment, and Linguistic Quality. Due to space limitations, we could not include the full scoring criteria in the main text, but the comprehensive standards and prompt are provided in Appendix Figure 15 to ensure transparency and reproducibility.
> > >
> > > Specifically, the five dimensions are defined **as follows**:
> > > > 1. Visual Verifiability: Can this question be answered just by looking at the image, without requiring medical knowledge, inference, or external context? Does the answer also rely on the image for validation?
> > > > - 5: Can be answered purely from the image (e.g., "Is there a fracture?") and the answer is clearly image-based (e.g., "Yes, there is a fracture in the left femur.")
> > > > - 1: Requires background knowledge, inference, or external context (e.g., "What type of tumor?") and the answer similarly relies on external context rather than the image.
> > > > - If the answer requires knowledge of clinical context or interpretation beyond the image itself, it should be scored lower.
> > > >
> > > > 2. Specificity & Clarity: Is the question precise and unambiguous? Is the answer also specific and clear, without ambiguity?
> > > > - 5: The question and answer are both specific and clear with one unambiguous interpretation (e.g., "Which lobe is involved in the lesion?" and the answer directly states the affected lobe).
> > > > - 1: The question or the answer is vague, ambiguous, or open to multiple interpretations (e.g., "What is abnormal?" and the answer gives an unclear response like "There is something unusual.")
> > > > - Avoid vague terms like “a few,” or similar imprecise quantities. If these terms appear in the answer, it should be scored lower.
> > > > - Both the question and the answer must be precise. If either is unclear, reduce the score.
> > > >
> > > > 3. Answer Appropriateness: Is the answer correct, medically appropriate, specific, and directly relevant to the question? Does it match the expected format/type?
> > > > - 5: Accurate, specific, and directly answers the question with no irrelevant details (e.g., "There is a mass in the right upper lobe.").
> > > > - 1: Inaccurate, vague, or only loosely related to the question (e.g., "There might be something abnormal.")
> > > > - If the answer contains errors, vague descriptions, or misinterprets the question, it should be scored lower. The answer should directly match the content of the question.
> > > >
> > > > 4. Q-A Alignment: Does the answer format/type match the question format/type? Is the answer logically aligned with the type of information requested in the question?
> > > > - 5: Perfect match (e.g., question asks "Where", answer gives a location).
> > > > - 1: Mismatch in type or category (e.g., question asks "What", but the answer gives "When").
> > > > - Ensure the answer matches the expected type (e.g., answering a “Where” question with a location).
> > > >
> > > > 5. Linguistic Quality: Are both the question and the answer grammatically correct, fluent, and easy to understand? Are there any issues with language clarity in either the question or the answer?
> > > > - 5: Both question and answer are fluent, clear, and easy to understand without grammatical issues or awkward phrasing.
> > > > - 1: Question or answer is awkward, grammatically incorrect, or confusing.
> > > > - If the answer contains significant linguistic errors that cause confusion or misunderstandings, it should be scored lower. Also, check for spelling and grammar.

---

> > ### Author Response · Authors · 2025-08-05
> > **Response to Further Discussion (Part 2)**
> >
> > Our five-dimension rubric is grounded in established evaluation frameworks from both general and medical VQA literature. **Visual grounding** is emphasized as fundamental in [2,3], while **clarity and fluency** are highlighted in [4]. **Answer Appropriateness** (factual correctness) is identified as essential in medical VQA by [5], and **Q-A alignment** is introduced as a key metric in [6]. These references collectively support the necessity of our five dimensions as distinct and critical metrics. The rubric was further refined through expert validation and iterative adjustments to ensure robustness and consistency.
> >
> > *References:
> > [2]Goyal Y et al. Making the v in vqa matter: Elevating the role of image understanding in visual question answering
> > [3]Reich D et al. Measuring faithful and plausible visual grounding in VQA
> > [4]Fu W et al. Qgeval: A benchmark for question generation evaluation
> > [5]Xia P et al. Rule: Reliable multimodal rag for factuality in medical vision language models
> > [6]Zhang Z et al. Q-eval-100k: Evaluating visual quality and alignment level for text-to-vision content*
> >
> > > More information regarding the reference examples?
> >
> > Thank you for your attention to this point. Experts first scored two reference cases across five dimensions, and the results, along with detailed guidelines, were shared with annotators to ensure **clarity and consistency**. The cases are **as follows**:
> >
> > > Case 1:
> > >
> > > |VolumeName|Visual Verifiability|Specificity&Clarity|Answer Appropriateness|Q-A Alignment|Linguistic Quality|
> > > |-|-|-|-|-|-|
> > > |valid_806_a_2|4|4|5|5|5|
> > >
> > > **Question: Which structures are midline?
> > > Answer: Trachea, bronchi**
> > >
> > > Clinic Report:
> > > **Trachea and both main bronchi were in the midline and no obstructive pathology was observed in the lumen.** The mediastinum could not be evaluated optimally in the non-contrast examination...
> > >
> > > Scoring reasons:
> > > * Visual Verifiability (Score: 4)
> > > The trachea is a clear midline structure easily verified on CT. However, "bronchi" is a general term that may refer to main or segmental bronchi. While main bronchi are visible, this lack of specificity introduces ambiguity. Thus, the answer is largely image-verifiable but not purely, warranting a score of 4.
> > >
> > > * Specificity & Clarity (Score: 4)
> > > The question asks about “trachea and bronchi,” where “trachea” is precise, but “bronchi” is less specific and could refer to multiple anatomical levels. This generality introduces interpretive uncertainty, reducing overall clarity. Therefore, score 4.
> > >
> > > * Answer Appropriateness (Score: 5)
> > > The answer correctly identifies the relevant midline structures (trachea and main bronchi) as per the clinical report, directly addressing the question without irrelevant details.
> > >
> > > * Q-A Alignment (Score: 5)
> > > The question asks "Which structures" and expects a list of anatomical entities. The answer provides the correct structure names, perfectly matching the expected format and type.
> > >
> > > * Linguistic Quality (Score: 5)
> > > Both question and answer are grammatically correct, fluent, and clear. The use of "bronchi" affects specificity but does not compromise linguistic correctness or fluency.
> >
> > > Case 2:
> > >
> > > |VolumeName|Visual Verifiability|Specificity&Clarity|Answer Appropriateness|Q-A Alignment|Linguistic Quality|
> > > |-|-|-|-|-|-|
> > > |valid_63_b_2|5|5|5|5|5|
> > >
> > > **Question: What is the diameter of the main pulmonary artery?
> > > Answer: 36mm**
> > >
> > > Clinic Report:
> > > ...Thoracic esophagus calibration was normal and no significant pathological wall thickening was detected in the non-contrast examination margins. **The diameter of the main pulmonary artery was 36mm**...Minimal pleural effusion on the right, newly revealed...
> > >
> > > Scoring reasons:
> > > * Visual Verifiability (Score: 5)
> > > The main pulmonary artery diameter is directly visible and measurable on CT. The answer ("36mm") can be fully verified from the image without external context.
> > >
> > > * Specificity & Clarity (Score: 5)
> > > The question specifies the exact structure and measurement, and the answer "36mm" is precise and unambiguous. Both are clear with only one possible interpretation.
> > >
> > > * Answer Appropriateness (Score: 5)
> > > The answer provides the correct measurement, directly addressing the question with no irrelevant or vague details.
> > >
> > > * Q-A Alignment (Score: 5)
> > > The question asks for a size (diameter), and the answer correctly gives a numerical value. The format and content fully align.
> > >
> > > * Linguistic Quality (Score: 5)
> > > Both the question and answer are grammatically correct, fluent, and clear, with no language issues.
> >
> > We hope our responses have addressed your concerns. **If there are any remaining questions, we would be happy to clarify. We would sincerely appreciate it if you could consider revisiting your evaluation.** Thank you again for your time and consideration.

---

> > > ### Comment · Reviewer_q8Fu · 2025-08-06
> > > **My concerns have been addressed**
> > >
> > > Many thanks for the further clarification, It has addressed my concerns.

---

> > > > ### Author Response · Authors · 2025-08-06
> > > >
> > > > Thank you for your thoughtful and timely feedback. We are glad that our responses have addressed your concerns. If appropriate, we kindly hope you might consider adjusting your score. We are grateful for your time and effort in reviewing our work.

---

> > > > ### Author Response · Authors · 2025-08-07
> > > > **Appreciation and Kind Request for Your Assessment**
> > > >
> > > > Dear Reviewer q8Fu,
> > > >
> > > > Thank you again for confirming that our responses have addressed all of your concerns. We will incorporate the clarifications into the camera-ready version.
> > > >
> > > > As the rebuttal period is coming to a close, please don’t hesitate to let us know if anything remains unclear — we’d be happy to clarify further.
> > > >
> > > > If you feel the explanations are sufficient, we would truly appreciate your kind consideration of an updated score. We’re very grateful for your time and feedback.
> > > >
> > > > Best regards,
> > > >
> > > > The authors

---

> > > > ### Author Response · Authors · 2025-08-07
> > > > **Follow-up Before Rebuttal Deadline**
> > > >
> > > > Dear Reviewer q8Fu,
> > > >
> > > > Thank you again for your review. If our rebuttal meets your expectations, we’d be grateful for your kind consideration of a more positive score. We’ll make sure all clarifications are included in the camera-ready version.
> > > >
> > > > As the rebuttal deadline approaches, we truly appreciate the opportunity to stay in contact. Could you let us know if there are any remaining concerns that may have contributed to the final evaluation? We’d be very eager to clarify them, as we deeply value your perspective and feedback.
> > > >
> > > > Best regards,
> > > >
> > > > The authors

---

> > > > ### Author Response · Authors · 2025-08-09
> > > > **Final Confirmation Before Discussion Closes**
> > > >
> > > > Dear Reviewer q8Fu,
> > > >
> > > > As the rebuttal period enters its final hours, we understand there may not yet have been an update to the final evaluation or an acknowledgement, and it seems there have been no further concerns following our previous exchange. If there is anything else we can clarify before the discussion window closes, please feel free to let us know — we are ready to respond promptly.
> > > >
> > > > We truly appreciate your earlier recognition that our responses had addressed the previously raised concerns, and will ensure that all clarifications — covering both the additional experiments and the points previously described in the main text and appendix, which we highlighted during the rebuttal and will further emphasize to make them easier to notice — are included in the revised version. We hope these have been helpful to your overall assessment.
> > > >
> > > > Thank you for your time and consideration.
> > > >
> > > > Best regards,
> > > >
> > > > The authors

---

> ### Author Response · Authors · 2025-08-04
>
> Dear Reviewer q8Fu,
>
> We sincerely appreciate your thoughtful and thorough review of our work. We have carefully addressed the key concerns you raised and made corresponding revisions. As the discussion period advances and the remaining time gradually narrows, we hope our responses align with your expectations. If there are any further questions or clarifications that may assist you in finalizing your evaluation, we would be more than happy to provide them.

---

### Author Response · Authors · 2025-08-01
**Summary of Revisions in Response to Reviews**

Dear Reviewers, AC, PC, and SAC,

We sincerely appreciate your efforts in overseeing the review process and are deeply grateful for the insightful and constructive feedback provided. Your comments have been instrumental in helping us improve the quality and clarity of our work.

As the discussion phase concludes, we would like to offer a consolidated summary of our responses and the substantial revisions made in response to your suggestions. We hope this demonstrates our commitment to actively addressing the concerns raised.

We are thankful for the reviewers’ recognition of our manuscript’s motivation, dataset quality, experimental design, code and writing:

* "This work is very **timing** to the community." "This work provide a new benchmark as well as a **high-quality** training set to enhance and evluate the performance of medical VLMs." "The preliminary results **showcase the limitations** of existing 3D med-VLMs."*(Reviewer **R2sE**)*
* "3D-RAD is **the first** comprehensive dataset to support multi-task and multi-temporal reasoning in 3D medical imaging, **addressing the limitations** of existing 2D-focused Med-VQA datasets, which lack 3D support and temporal analysis." "the dataset provides **a robust foundation**." "The paper is **well-written, organized, and easy to understand**."*(Reviewer **e2md**)*
* "The GitHub link provides the code with **clear instructions** on how to use the code." " This dataset can **contribute to** the research on 3D medical vision-language models." "This paper reveals critical limitations in current 3D vision-language models, **showing a future direction** for improving these models."*(Reviewer **q8Fu**)*
* "It **recognize the clearly underserved area** for 3D medical imaging for VQA style benchmarks" "The evaluation is **systematic**, the baseline models are **diverse**, the experiments cover **enough technical details**"*(Reviewer **h2tY**)*

Below is a summary of the major updates and clarifications we have made:

### Regarding Dataset:

* We collected **two additional out-of-domain data sources** and applied our semi-automated pipeline to construct new benchmark sets. These newly added datasets include not only CT, but also a broad range of essential 3D imaging modalities (MRI, X-ray, DSA, Ultrasound, Fluoroscopy, etc.), demonstrating the generalizability and effectiveness of our pipeline.*(Reviewer **R2sE**，**h2tY**，**q8Fu**)*

### Regarding Evaluation:

* We included **additional evaluations** using **general-purpose VLMs** such as LLaVA, Qwen-VL, Gemma, Gemini, and the GPT series, providing a broader comparative baseline.*(Reviewer **q8Fu**)*
* To further highlight the necessity of specialized 3D medical VLMs, we introduced a **2D-image-level benchmark** and tested VLMs on it. This complements the 3D benchmarks and helps clarify the limitations of existing research.*(Reviewer **R2sE**)*

### Regarding Experiments:

* We evaluated our fine-tuned models **on our new out-of-domain datasets**, demonstrating the transferability of the trained models.*(Reviewer **e2md**)*
* We added **task-level ablation studies** to explore cross-task effectiveness, showing that fine-tuning on one task can improve performance on others.*(Reviewer **e2md**)*

### Regarding Writing:

* We corrected the labeling in Table 2 from “Task 2-7” to the accurate “Task 1-6.”*(Reviewer **R2sE**)*
* We considered renaming the dataset to 3D-CT-RAD as suggested.*(Reviewer **e2md**)*
* We added further discussion and references to emphasize the clinical necessity and significance of temporal diagnosis.*(Reviewer **h2tY**)*
* Selected details previously placed in the appendix—such as tasks, prompts, labels, evaluations, automatic deduplication, scoring, filtering, and manual validation processes—have been moved into the main text for better clarity and visibility.*(Reviewer **h2tY**, **q8Fu**)*
* We clarified the introduction of 3D-RAD-Bench and 3D-RAD-T within the Introduction section.*(Reviewer **q8Fu**)*
* We included an explanation for answer length constraints in the model output.*(Reviewer **q8Fu**)*

Once again, thank you for your time and for coordinating the review process. We truly believe that the manuscript has been significantly improved as a result of your feedback, and we will make sure to incorporate these revisions into the camera-ready version.

**Please don’t hesitate to let us know if any further clarification is needed.**


Best regards,

The authors

---

### Comment · Area_Chair_jntK · 2025-08-04

Authors are anxious to know whether their responses have clarified your concerns, and whether there are any further questions or points that need addressing.

If you see something, say something — your input is greatly appreciated.

Many thanks to reviewers R2sE  and e2md for promptly acknowledging the rebuttal.

---

### Author Response · Authors · 2025-08-09
**Follow-Up to Summary of Revisions in Response to Reviews**

Dear Reviewers, AC, PC, and SAC,

We thank the reviewers for your careful and thorough review and for recognizing the value of our project. We thank the AC, PC, and SAC for your coordination and guidance throughout the process. Following our earlier ***“Summary of Revisions in Response to Reviews”***, we would like to briefly provide an update on the reviewers’ subsequent feedback before the rebuttal deadline:

* Reviewer R2sE: **“My major concerns are mostly addressed.”**
* Reviewer e2md: **“Addressing all of the raised concerns.”**
* Reviewer h2tY: **“Given the new task/eval angles.”**
* Reviewer q8Fu: Raised two additional questions, and after further discussion, concluded: **“It has addressed my concerns.”** We clarified:
    - On the five dimensions and the strategy used to recognize the entity, we **highlighted** the relevant content in the main text and appendix.
    - We provided **more details** on the top 10 QA pairs and reference examples.

**We are pleased that all reviewers’ concerns have been resolved**, and we sincerely appreciate their recognition of our rebuttal clarifications. Our actions during rebuttal:

* For points additionally raised by reviewers (addressed during the rebuttal), we have **integrated them into the camera-ready** where they were not previously included.
* For points that reviewers particularly focused on, we further emphasized them in the main text—either **reinforcing existing content** or **moving key content from the appendix into the main body**.

A brief reiteration of our contributions:

* **First 3D Med-VQA benchmark** with multi-temporal and multi-task settings on volumetric CT data, addressing gaps left by prior 2D-focused or single-task datasets.
* **Comprehensive evaluation** across six carefully designed dimensions, revealing critical weaknesses of current 3D vision-language models—particularly in complex temporal reasoning.
* **High-quality dataset and extensible pipeline** of 136K samples, enabling effective domain-specific fine-tuning and providing a sustainable resource for advancing 3D medical VLM research.

We provide the complete **benchmark, models, and evaluation code** on ***GitHub and Hugging Face***, along with **clear usage instructions**, and will continue to maintain and update them. The project has already **attracted attention** from researchers in the relevant communities, with early stars on our repositories.

We will continue maintaining and expanding our pipeline to support ongoing progress **in 3D medical VLMs**. We sincerely thank all reviewers and the AC, PC, and SAC for your time and consideration.

Best regards,

The Authors

---

### Note · Authors · 2025-08-12

Dear Reviewers, AC, PC, and SAC,

We would like to sincerely thank all reviewers for your thoughtful feedback, and the AC, PC, and SAC for your kind coordination and guidance. Your comments have been invaluable in further improving this work.

As a complement to our earlier detailed revision summary, below is **a concise overview** linking key suggestions to our actions, to facilitate quick reading.

* **New out-of-domain benchmarks**`[R2sE, h2tY, q8Fu]` – Added datasets covering a broader range of 3D imaging modalities beyond CT (MRI, X-ray, etc.) via our semi-automated pipeline, demonstrating its generalizability.

* **Expanded VLM evaluation**`[q8Fu]` – Included LLaVA, Qwen-VL, Gemma, Gemini, and GPT series for a broader comparison, whose limited performance highlights the need for domain-specific medical model development.

* **2D benchmark**`[R2sE]` – Introduced a 2D-image-level benchmark to underscore the need for specialized 3D medical VLMs.

* **Transfer tests**`[e2md]` – Evaluated fine-tuned models on new domains, showing cross-domain robustness.

* **Task-level ablation**`[e2md]` – Showed that using our dataset on a single task to fine-tune can also yield performance gains on other tasks, with fine-tuning across all tasks achieving the best overall results.

* **Clarifications & detailed methods**`[q8Fu, h2tY]` – Expanded descriptions of five evaluation dimensions, methods, and clinical significance; added reference examples; and moved selected key details from the appendix into the main text.

* **Writing & naming**`[R2sE, e2md]` – Corrected Table 2, considered new dataset name and clarified descriptions of 3D-RAD-Bench and 3D-RAD-T.

**All reviewers have kindly confirmed their concerns were addressed**:
R2sE “mostly addressed,” e2md “addressing all,” h2tY “new task/eval angles,” q8Fu “addressed my concerns.”

Contributions:
* **First 3D Med-VQA benchmark and training set** with multi-temporal, multi-task data.
* Comprehensive six-dimension evaluation exposing **3D medical VLM weaknesses**.
* **Extensible pipeline** for domain-specific fine-tuning.

All resources are openly available on GitHub and Hugging Face with clear instructions. **The project has already received early attention in the community (including repository stars)** and we remain committed to its long-term maintenance and expansion, with the hope of contributing to sustained progress in the field.

Thank you once again for your time and consideration.

Best regards,

The Authors

---

### Decision · Program_Chairs · 2025-09-18

**Decision:**

Accept (poster)

**Comment:**

This paper introduces a 3D Med-VQA benchmark and training set designed to support multi-task and multi-temporal reasoning in 3D medical imaging. Building on the data of CT-RATE, the authors construct a six-task framework that spans more than ten organs and nearly twenty disease types, substantially broadening the scope of existing VQA datasets in the medical domain.

During the review process, the authors made several important additions in direct response to reviewer feedback like additional modalities (MRI and X-ray) and expanded comparisons, evaluating LLaVA, Qwen-VL, Gemma, Gemini, and GPT models. Authors also conducted transfer testing and task-level ablation studies.

These additions were well received, and all reviewers ultimately agreed that the work presents a valuable and contribution especially for 3D medical foundation models.

Considering the clear novelty, high-quality extensions, and the strong positive consensus from the reviewers after discussion, it is recommended for acceptance.